# Axon TRAP reveals learning-associated alterations in cortical axonal mRNAs in the lateral amygdala

Linnaea E Ostroff[1]*, Emanuela Santini[2†], Robert Sears[3,4,5†], Zachary Deane[1], Rahul N Kanadia[1], Joseph E LeDoux[3,4], Tenzin Lhakhang[6], Aristotelis Tsirigos[6,7], Adriana Heguy[7,8], Eric Klann[3]*

[1]Department of Physiology and Neurobiology, University of Connecticut, Storrs, United States; [2]Department of Neuroscience, Karolinska Institutet, Solna, Sweden; [3]Center for Neural Science, New York University, New York, United States; [4]Emotional Brain Institute, Nathan Kline Institute for Psychiatry Research, Orangeburg, United States; [5]Department of Child and Adolescent Psychiatry, New York University School of Medicine, New York, United States; [6]Applied Bioinformatics Laboratories, New York University School of Medicine, New York, United States; [7]Department of Pathology, New York University School of Medicine, New York, United States; [8]Genome Technology Center, New York University School of Medicine, New York, United States

**Abstract** Local translation can support memory consolidation by supplying new proteins to synapses undergoing plasticity. Translation in adult forebrain dendrites is an established mechanism of synaptic plasticity and is regulated by learning, yet there is no evidence for learning-regulated protein synthesis in adult forebrain axons, which have traditionally been believed to be incapable of translation. Here, we show that axons in the adult rat amygdala contain translation machinery, and use translating ribosome affinity purification (TRAP) with RNASeq to identify mRNAs in cortical axons projecting to the amygdala, over 1200 of which were regulated during consolidation of associative memory. Mitochondrial and translation-related genes were upregulated, whereas synaptic, cytoskeletal, and myelin-related genes were downregulated; the opposite effects were observed in the cortex. Our results demonstrate that axonal translation occurs in the adult forebrain and is altered after learning, supporting the likelihood that local translation is more a rule than an exception in neuronal processes.

*For correspondence:
linnaea.ostroff@uconn.edu (LEO);
ek65@nyu.edu (EK)

†These authors contributed equally to this work

Competing interests: The authors declare that no competing interests exist.

## Introduction

Neurons use local translation as a means of rapid, spatially restricted protein regulation in their distal processes, particularly during remodeling driven by external cues (*Donnelly et al., 2010*; *Wang et al., 2010*; *Holt and Schuman, 2013*). Memory consolidation requires new proteins to stabilize molecular changes induced by learning (*Davis and Squire, 1984*; *Mayford et al., 2012*), and local translation in dendrites is thought to be an essential source of these proteins (*Sutton and Schuman, 2006*). Rich and diverse assortments of mRNAs have been described in neuropil of the mature hippocampus (*Poon et al., 2006*; *Zhong et al., 2006*; *Cajigas et al., 2012*) and in cortical synaptoneurosomes (*Ouwenga et al., 2017*), underscoring the importance of decentralized translation in synaptic function. Yet no role for axonal translation in learning and memory has been reported in the adult forebrain.

Translation has long been known to occur in invertebrate axons, and it is now established to be essential for growth and response to guidance cues in developing CNS axons, and in regeneration of PNS axons (*Akins et al., 2009*; *Twiss and Fainzilber, 2009*; *Jung et al., 2012*; *Batista and Hengst, 2016*). Adult forebrain axons, in contrast, traditionally have been characterized as lacking the capacity for translation, in part due to a lack of reliable evidence, and in part to the perception that they are structurally and functionally inert compared to dendrites and immature axons (*Kindler et al., 2005*; *Jung et al., 2012*; *Batista and Hengst, 2016*). However, a number of recent studies have shown that mature axons are in fact capable of translation, at least in some circumstances (*Willis et al., 2007*; *Gumy et al., 2011*; *Kalinski et al., 2015*), including in the CNS (*Taylor et al., 2009*; *Baleriola et al., 2014*; *Kar et al., 2014*; *Shigeoka et al., 2016*; *Hafner et al., 2019*). This work has largely been done with cultured neurons, but one study used translating ribosome affinity purification (TRAP) to isolate ribosome-bound mRNAs in retinal ganglion cells (RGCs) of adult mice (*Shigeoka et al., 2016*), demonstrating that ribosome-bound mRNAs are present in adult CNS axons in vivo. Presynaptic translation has been shown to be necessary for long-term depression in hippocampal (*Younts et al., 2016*) and striatal (*Yin et al., 2006*) slice preparations from young animals, indicating that axonal translation is involved in synaptic plasticity and therefore could be important in memory as well.

Aversive auditory Pavlovian conditioning (fear or threat conditioning), in which animals learn to associate an auditory tone with a foot shock, is supported by persistent strengthening of synaptic inputs to the lateral amygdala (LA) from auditory areas (*Johansen et al., 2011*). The LA receives strong excitatory input from auditory cortical area TE3 (*Romanski and LeDoux, 1993*; *Shi and Cassell, 1997*; *Farb and Ledoux, 1999*), and Pavlovian conditioning induces persistent enhancement of presynaptic function at these synapses (*McKernan and Shinnick-Gallagher, 1997*; *Tsvetkov et al., 2002*; *Humeau et al., 2003*; *Schroeder and Shinnick-Gallagher, 2005*). Consolidation of memory requires translation in the LA (*Schafe and LeDoux, 2000*), and we have found that it induces changes in the translational machinery in LA dendrites associated with synapse enlargement (*Ostroff et al., 2010*). Intriguingly, we also found that learning-induced structural changes occurred at individual axonal boutons as opposed to uniformly along axons, suggesting that plasticity may be as synapse-specific and compartmentalized on the presynaptic side as it is on the postsynaptic side (*Ostroff et al., 2012*). To determine whether axonal translation is involved in memory formation, we confirmed the presence of translation machinery in LA axons, and combined TRAP with RNAseq to identify changes in the translatome of auditory cortical axons during memory consolidation.

## Adult axons contain translation machinery

Early electron microscopy studies reported abundant polyribosomes in the somata and dendrites of neurons, but rarely in axons (reviewed by *Giuditta et al., 2008* and *Jung et al., 2012*). However, the paucity of conspicuous polyribosomes does not necessarily preclude translation. Regenerating sciatic nerve axons contain mRNAs and translate membrane proteins in vivo, but do not show ultrastructural evidence of polyribosomes or rough endoplasmic reticulum (*Zheng et al., 2001*; *Merianda and Twiss, 2013*). In addition, hippocampal interneuron axons contain ribosomal proteins (*Younts et al., 2016*). This suggests that translation sites other than the classic morphological structures do exist, such as the periaxoplasmic ribosomal plaques found in adult spinal cord axons (*Koenig, 2009*). Recent work in yeast has shown that translation can occur on 80S monosomes, with a bias toward highly regulated transcripts (*Heyer and Moore, 2016*).

We have occasionally observed polyribosomes in presynaptic boutons in the adult rat LA by EM (*Figure 1a–b*, *Figure 1—figure supplement 1a–e*), although these are infrequent (LO, unpublished observations). A possible explanation for this is that these axons contain translation machinery that does not usually assemble into polyribosomal structures with traditionally recognizable morphology, such as monosomes, whose morphology is not distinctive enough for unequivocal identification. To more directly assess the potential for translation in LA axons, we used immuno-electron microscopy to localize components of the translation machinery. Because translation initiation is most extensively regulated step in gene expression, as well as a critical mediator of memory formation (*Santini et al., 2014*), we focused on translation initiation factors. The eukaryotic initiation factors eIF4E, eIF4G, and eIF2α each were present in axons forming synapses onto spiny dendrites in the caudal dorsolateral subdivision of the LA (*Figure 1c–e*), which receives the most robust projections from TE3 (*Romanski and LeDoux, 1993*; *Shi and Cassell, 1997*; *Farb and Ledoux, 1999*), as was ribosomal

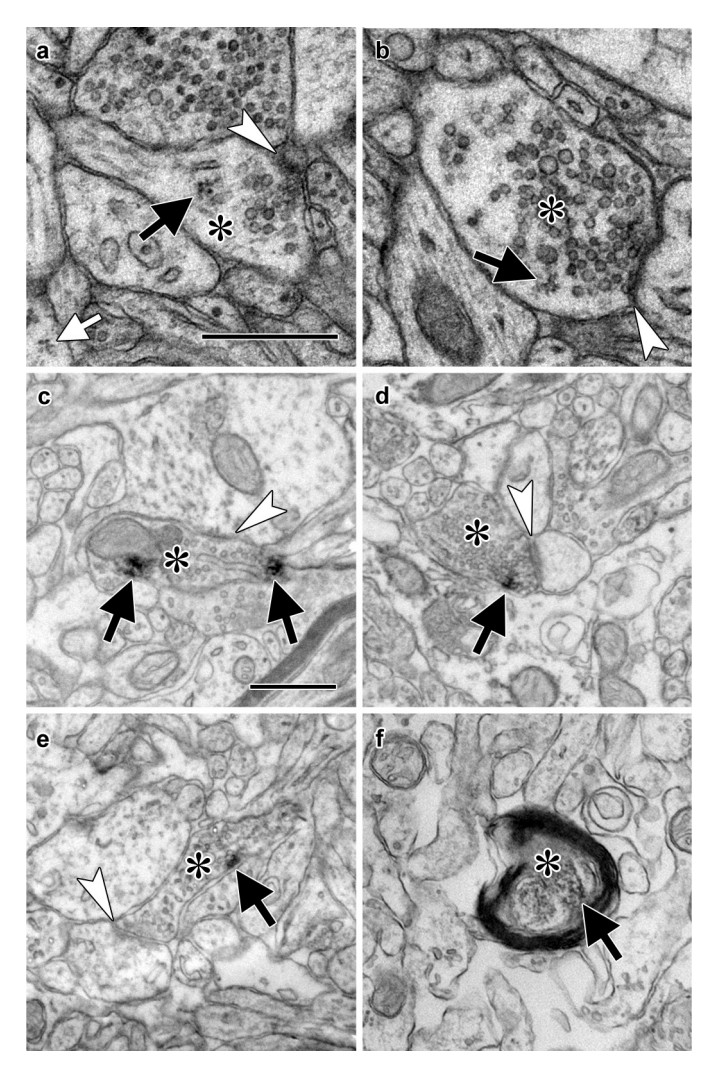

**Figure 1.** Electron micrographs of translation machinery in lateral amygdala axons. (**a–b**) Polyribosomes (black arrows) in axonal boutons (asterisks). A polyribosome in an astrocytic process (white arrow) is visible at the lower left of panel (**a**). (**c–e**) Axonal boutons (asterisks) containing immunolabeling (black arrows) for eIF4E (**c**), eIF4G1 (**d**), and eIF2$\alpha$ (**e**). White arrowheads indicate asymmetric synapses onto dendritic spines (a, d, and e) and shafts (**b** and **c**). (**f**) Myelinated axon (asterisk) containing immunolabel for ribosomal protein s6 (arrow). Scale bars = 500 nm. The online version of this article includes the following figure supplement(s) for figure 1:

**Figure supplement 1.** Polyribosomes and translation factors in axons.

protein S6 (*Figure 1f*). These synapses have the same classic excitatory morphology as the glutamatergic projections from TE3 to LA (*Farb and Ledoux, 1999*), consistent with local translation on TE3 inputs. Quantification of eIF4E immunolabel through serial sections of neuropil revealed that 63% of axons were labeled, along with 39% of dendritic spines and 100% of dendritic shafts (*Figure 1—figure supplement 1f–i*). Consistent with this pattern, we have previously found polyribosomes throughout dendritic shafts but in only a subset of dendritic spines, where their presence is regulated by learning (*Ostroff et al., 2010*).

## Isolation of the adult axonal translatome

To identify ribosome-bound mRNA transcripts in distal TE3 axons, we used TRAP (*Heiman et al., 2008*), in which a tagged ribosomal protein is expressed in cells of interest and used to immunoprecipitate ribosome-bound mRNA. A recent study used an HA-tagged ribosomal protein to examine

the translatome of retinal ganglion cell axons in both immature and adult mice (*Shigeoka et al., 2016*), and an eGFP-tagged ribosomal protein expressed in adult mouse layer V cortical neurons was observed in axons of the corticospinal tract (*Walker et al., 2012*), demonstrating that this method is viable in at least two types of adult CNS neurons in vivo. We used a viral vector to express an eYFP-ribosomal protein L10a fusion protein (*Kratz et al., 2014*) in TE3 cells in adult rats (*Figure 2a–b*). Pilot experiments using an adeno-associated viral vector resulted in moderate to strong retrograde infection of cells in afferent areas. To ensure that no cell bodies outside of the injection site expressed the construct, we switched to a lentiviral vector, which did not result in retrograde infection. Lentivirus has also been shown to have preferential tropism for excitatory neurons over inhibitory neurons in the cortex (*Nathanson et al., 2009*), which is advantageous since TE3 projections to the amygdala are excitatory. Immuno-electron microscopy confirmed the presence of eYFP in LA axons (*Figure 2c–f*).

TRAP was combined with Pavlovian conditioning to determine how the axonal translatome changes during memory consolidation (*Figure 3a*). Animals expressing eYFP-L10a in TE3 were given

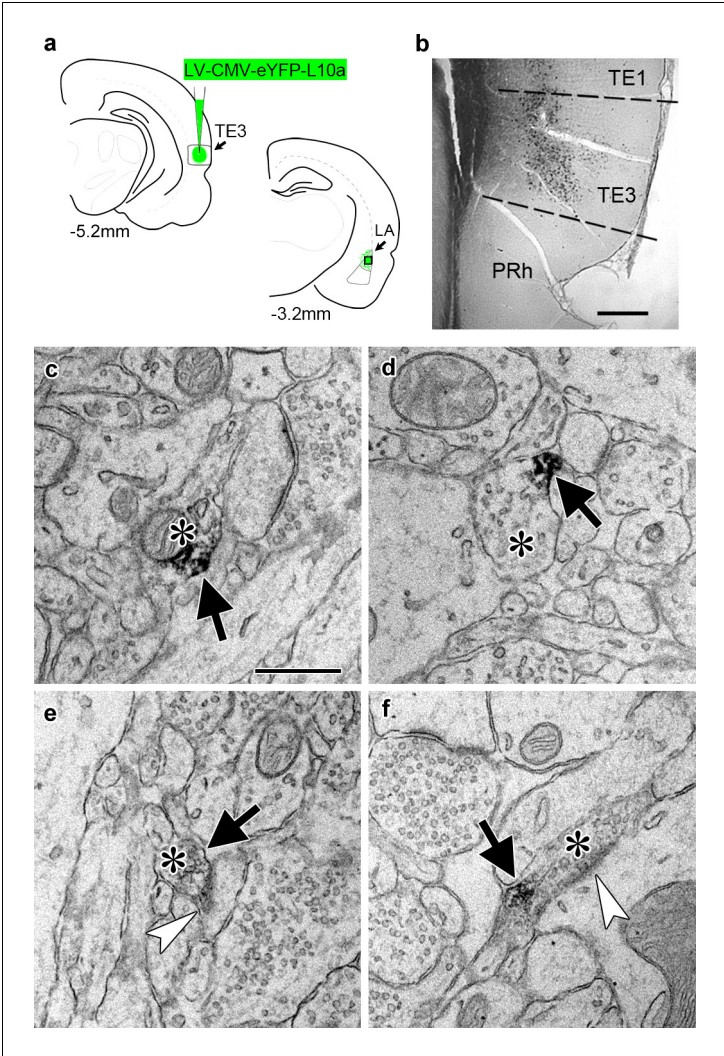

**Figure 2.** Transport of a tagged ribosomal L10a protein to cortical projection axons. (a) Schematic of injection site in cortical area TE3 and its lateral amygdala (LA) projection area, with AP coordinates from Bregma noted. The black square indicates the area of LA sampled for EM. PRh: perirhinal cortex. (b) Immunolabeling of YFP in transfected TE3. (c–f) Electron micrographs of LA showing axonal boutons (asterisks) containing YFP immunolabel (black arrows). The boutons in (e) and (f) are forming asymmetric synapses (white arrowheads) on a dendritic spine head (e) and a dendritic shaft (f). Scale bars = 500 μm in (b) and 500 nm in (c–f).

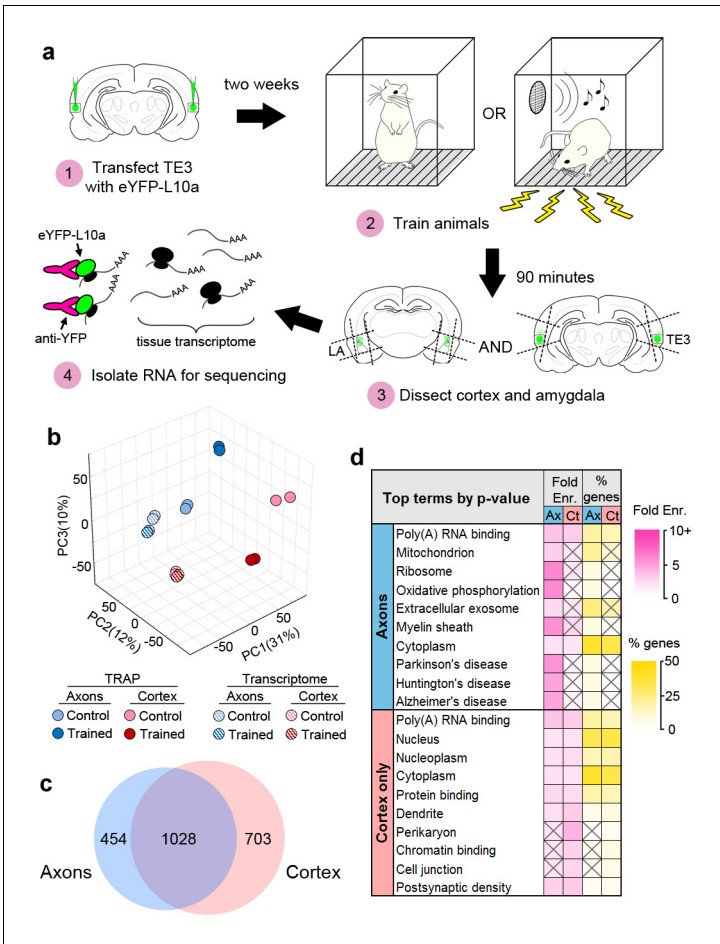

**Figure 3.** Isolation of the TE3 axonal translatome. (**a**) Experimental workflow (see text). (**b**) Principal component analysis of all experimental replicates. (**c**) Overlap between axonal and cortical translatomes. (**d**) Most enriched GO terms and KEGG pathways in axonal and cortex-only translatomes, sorted by Benjamini-Hochberg adjusted p-value. Gray X's indicate effects that were not significant (adjusted p-value>0.05).

The online version of this article includes the following figure supplement(s) for figure 3:

**Figure supplement 1.** Collection of TRAP samples.
**Figure supplement 2.** Filtering of DGE results.
**Figure supplement 3.** Comparison of TRAP and YFP-IP experiments.
**Figure supplement 4.** Composition of the axonal translatome.

either Pavlovian conditioning, consisting of auditory tones paired with mild foot shocks in a familiar chamber (the trained group), or exposure to the chamber alone (the control group). We did not present unpaired tones and shocks to the control group because this paradigm constitutes a different type of associative learning and results in plasticity at LA synapses (*Rogan et al., 2005*; *Ostroff et al., 2010*). Long-term memory formation requires de novo translation during a critical period of several hours after training (*Davis and Squire, 1984*; *Schafe and LeDoux, 2000*), thus we sacrificed animals during this time window and collected separate tissue blocks containing either the auditory cortex or the amygdala (*Figure 3—figure supplement 1a*). Although we refer to these samples as cortex and axons, the cortex samples also contain the proximal axon segments, myelinated segments that pass through the dorsal portion of the external capsule, as well as intrinsic projections and corticocortical projections terminating in adjacent areas of TE1 and perirhinal cortex (*Romanski and LeDoux, 1993*; *Shi and Cassell, 1997*). We should also note that our use of the term 'translatome' refers simply to the set of mRNAs that are bound to ribosomes, and therefore past the initiation step of translation, but these transcripts are not necessarily undergoing active elongation or termination at the moment of capture.

RNASeq was performed on the TRAPed mRNAs as well as the total mRNA isolated from the homogenized tissue blocks (the tissue transcriptome). Quality control metrics are shown in *Supplementary file 1*. Principal component analysis revealed correspondence between experimental replicates, as well as separation between the TRAPed samples and the transcriptome, the cortex and axons, and the trained and control groups (*Figure 3b*). Gene expression levels were correlated between replicates (*Figure 3—figure supplement 1b*). Differential gene expression (DGE) analysis was used to compare the eight groups (TRAPed mRNAs and the input tissue transcriptome from two brain areas of each of the two behavioral groups). Three types of comparisons were performed: TRAPed mRNAs were compared to the corresponding input tissue transcriptome, the axons and cortex were compared in each behavior group, and the behavior groups were compared in each brain area (*Supplementary file 2*). Comparison with a cell-type-specific proteome (*Sharma et al., 2015*) revealed that neuronal genes were more likely than non-neuronal genes to be enriched in the TRAPed samples versus the tissue transcriptome, whereas non-neuronal genes were more likely to be depleted (*Figure 3—figure supplement 1c*), confirming that our TRAPed samples contain mainly neuronal genes.

Because no translatome or transcriptome of adult forebrain axons has been previously published, we chose to take a conservative approach to defining axonal genes in our dataset (*Figure 3—figure supplement 2a*). In order to minimize false positives introduced by the TRAP procedure, only genes that were differentially expressed between TRAPed samples were included. Although this should account for much of the background from the experimental procedures, it does not account for differences between the background transcriptome of the tissue samples, and we therefore excluded genes that were differentially expressed in the corresponding tissue transcriptomes. Finally, genes that were differentially expressed between TRAPed samples were excluded if the enriched sample also was not enriched versus the tissue transcriptome. We defined genes that met these criteria as axonal if they were regulated by learning in the axons, enriched in the axons versus the cortex in either experimental group, or both. Examination of expression levels showed that our filtering method selected for more abundant genes with higher correlation between experimental replicates (*Figure 3—figure supplement 2b*). Of the 1482 axonal genes identified, the majority (1028) were also either regulated or enriched in the cortex (*Figure 3c*), and an additional 703 genes were regulated or enriched only in the cortex (defined as 'cortex-only' genes). It is important to note that although we are using the term 'translatome' to refer to the stringently selected subset of genes we used for analysis, the actual population of axonal mRNAs is almost certainly larger.

To directly assess the background introduced by the IP procedure, we repeated the TRAP experiment in animals that were not injected with the TRAP virus. In addition to the IP, mRNA binding to the overexpressed eYFP tag itself, as opposed to the tagged ribosomes, is another potential source of background. Instead of using an empty AAV backbone or a different reporter as a control, we used a lentivirus encoding eYFP to account for this possibility. As expected, there was substantial overlap between genes enriched in the TRAP and eYFP-IP samples versus the tissue transcriptome (*Figure 3—figure supplement 2c*). There were, however, very few learning-associated mRNAs in the eYFP-IP experiment, and these had little overlap with the TRAPed mRNAs, and even less after the filtering step. Although there was 47% overlap between axonal and cortical genes in the TRAP experiment (*Figure 3c*), there was only 2.5% overlap in the eYFP-IP experiment. These data confirm that the results of our TRAP experiment are not due to background. Because the eYFP-IP experiment targeted axonal eYFP, these samples were likely enriched for axonal mRNAs, ribosome-bound or not. Our data cannot distinguish these, but the background levels of extra-axonal mRNA in our dataset may be even lower than this control experiment indicates.

## Composition of the axonal translatome

To characterize the axonal translatome, we used DAVID (*Huang et al., 2009*) (https://david.ncifcrf.gov, version 6.8) to identify Gene Ontology (GO) Terms and KEGG Pathways enriched in the axonal and cortex-only gene sets. Complete results of DAVID analyses are in *Supplementary file 4*. The most significantly enriched terms in axons related to mitochondria, translation, and neurodegenerative diseases, whereas cortex-only genes were enriched for terms associated with the cell body, nucleus, and dendrites (*Figure 3d*). To ensure that our filtering process did not dramatically skew the composition of the final dataset, we also analyzed the unfiltered set of axonal genes. The resulting list of terms was similar, although enrichment levels were lower, consistent with a lower signal-

to-noise ratio in the unfiltered data (*Figure 3—figure supplement 3a*). Comparison between the filtered data from the TRAP and eYFP-IP experiments revealed little similarity between the most enriched GO terms (*Figure 3—figure supplement 3b*). Manual grouping of significantly enriched terms revealed that terms relating to the presynaptic compartment and cytoskeleton were also predominantly found in axons, along with terms relating to various other cellular functions such as the ubiquitin-proteasome pathway, GTPase signaling, and intracellular transport (*Figure 3—figure supplement 4a*).

The size and composition of the TE3 axonal translatome are similar to what has been reported in the translatomes of retinal ganglion cell axons (*Shigeoka et al., 2016*) and cortical synaptoneurosomes (*Ouwenga et al., 2017*), the transcriptome of adult hippocampal neuropil (*Poon et al., 2006*; *Zhong et al., 2006*; *Cajigas et al., 2012*), and the transcriptomes of axons isolated from cultures of dorsal root ganglion (*Willis et al., 2007*; *Gumy et al., 2011*), cultured motor neurons (*Briese et al., 2016*), and mixed cortical/hippocampal neurons (*Taylor et al., 2009*). We compared these datasets to our axonal and cortex-only translatomes and found greater overlap with the axonal genes, with 904 of the 1482 genes (60%) present in at least one published dataset (*Figure 3—figure supplement 4b*). Given that these data were obtained from different cell types, preparations, ages, and species, this suggests that at least some aspects of the axonal transcriptome are universal. In particular, transcripts associated with protein synthesis and energy metabolism are found throughout the various datasets. Interestingly, our axonal translatome had substantially more overlap with datasets from immature versus mature axons, potentially reflecting recapitulation of developmental mechanisms in learning.

## Opposite changes after learning in axons and cortex

The majority of genes in the translatome (1647 of 2185 or 75%) showed differential expression following learning, with 19% (415) and 6% (123) of the remainder enriched in the cortex or axons, respectively. Of regulated genes, 40% showed significant changes in both axons and cortex, and all but one of these (the mitochondrial enzyme *Dlst*) were regulated in opposite directions (*Figure 4a*). The magnitude of change in the axons and cortex was significantly correlated for these genes, particularly for those downregulated in axons and upregulated in cortex (*Figure 4b*). Expression levels in the axons and cortex were significantly correlated in both training groups regardless of learning effects, although genes that were upregulated in the axons showed the highest correlation (*Figure 4—figure supplement 1a–b*). In the control group, genes that were downregulated in axons showed the lowest correlation between the two areas, but this increased in the trained group, particularly for genes that were also upregulated in the cortex. These results suggest that the axonal translatome is not regulated independently, but that compartment-specific translation is coordinated within the cell. This is underscored by the fact that only 63 genes encompassed the 50 most abundant in both areas and conditions (*Figure 4—figure supplement 1c*). Genes that were upregulated in axons had the highest expression levels in both areas and conditions, further suggesting common regulatory mechanisms (*Figure 4c*). In contrast to the TRAP experiment, there was no overlap between the 115 genes regulated after learning in axons and the 21 regulated in cortex in the eYFP-IP experiment.

Performing DAVID analysis separately on upregulated and downregulated genes revealed that learning was associated with inverse, function-specific changes in the axonal and cortical translatomes (*Figure 4d*). To further explore the learning-associated changes in cellular functions, we used Ingenuity Pathway Analysis (IPA) software (Qiagen). IPA evaluates changes in gene expression with respect to a database of known pathways and functions, and assigns an enrichment p-value along with a z-score predicting activation or inhibition of a pathway based on published data. A search for upstream regulators found that most of the enriched pathways had opposite z-scores in the axons and cortex (*Figure 4e*, *Supplementary file 6*). Analysis of functional annotations with IPA similarly revealed opposing functional regulation in the two areas (*Figure 4—figure supplement 2a*, *Supplementary file 7*). Although the axonal transcriptome is theoretically a subset of the somatic transcriptome, these results demonstrate an unexpected degree of coordination between the axonal and cortical translatomes.

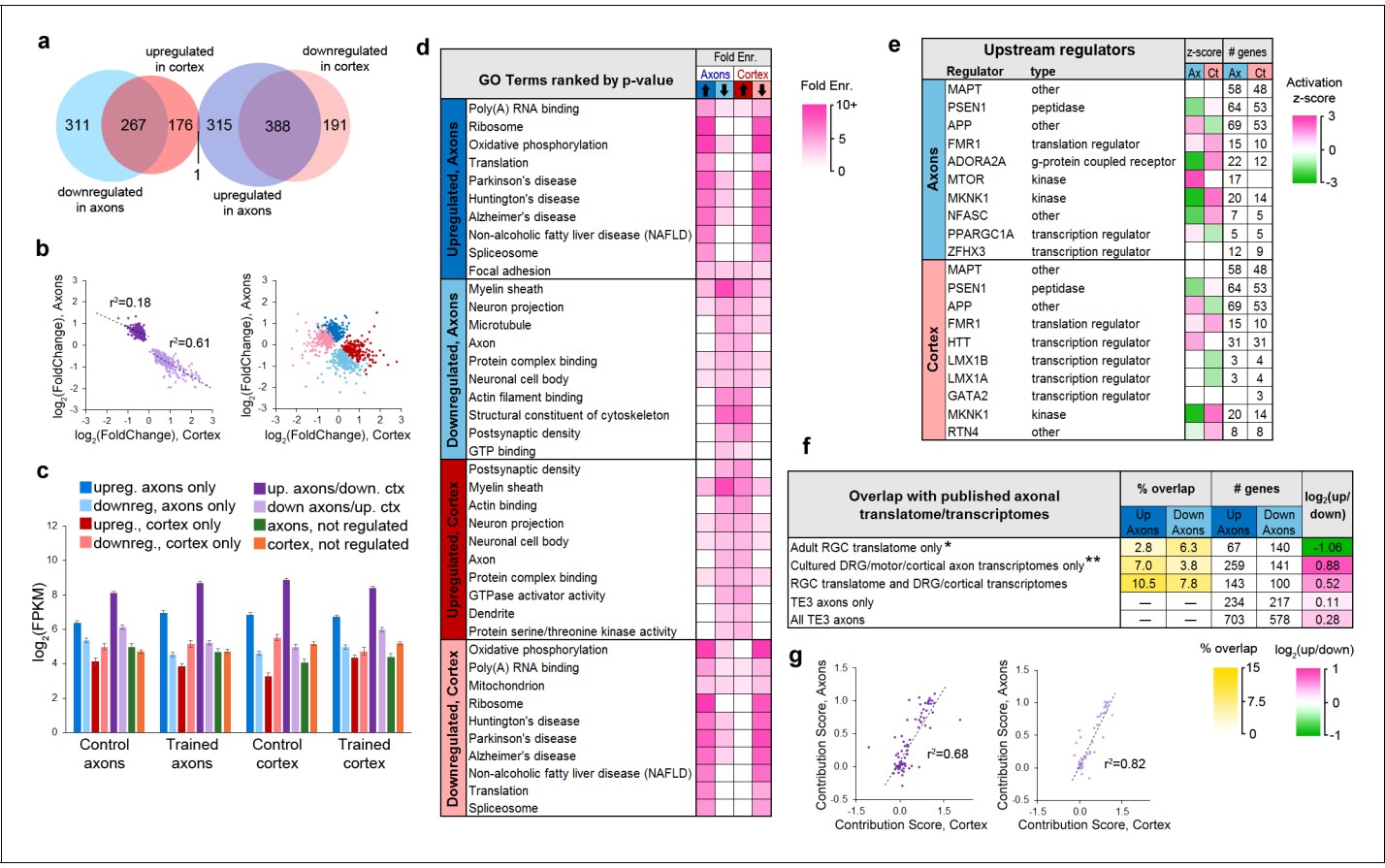

**Figure 4.** Learning-associated changes in the axonal translatome. (**a**) Overlap of training effects in the axons and cortex. (**b**) Correlations between effect sizes in the axons and cortex for genes differentially expressed in both areas after learning (left) or only one area (right). Regression lines are shown for correlations significant at $p < 1 \times 10^{-5}$. (**c**) Mean expression levels of genes in each group with respect to training effects. Results of ANOVA and post hoc test are given in *Supplementary file 5*. Error bars = s.e.m. (**d**) Top GO term and KEGG pathways enriched >3 fold in learning-regulated genes, ranked by Benjamini-Hochberg adjusted p-value. Highly redundant terms are not shown. (**e**) Top regulatory pathways affected by training in axons and cortex, sorted by adjusted p-value. Activation z-score represents the probability of a pathway being activated or inhibited after learning. (**f**) Overlap between genes up- or downregulated in axons by learning and published axonal translatomes and transcriptomes. * Data from *Shigeoka et al. (2016)*; ** Data from *Gumy et al. (2011)*, *Willis et al. (2007)*, *Taylor et al. (2009)*, and *Briese et al. (2016)*. (**g**) For genes that had multiple transcripts and were regulated by training in both axons and cortex, the contribution of each transcript to the gene-level effects in axons and cortex were correlated for genes upregulated in axons and downregulated in cortex (left) and genes downregulated in axons and upregulated in cortex (right). The contribution score was calculated as (change in FPKM transcript)/(change in FPKM gene).

The online version of this article includes the following figure supplement(s) for figure 4:

**Figure supplement 1.** Relative abundance of genes in axons and cortex.

**Figure supplement 2.** Ingenuity Pathway Analysis and comparison with published translatomes and transcriptomes.

**Figure supplement 3.** Transcript-level analysis.

## Learning-associated changes in the axonal translatome

Learning was associated with changes in genes related to a range of cellular processes, with some clear patterns of upregulation and downregulation. An overview of regulated genes is shown in *Table 1*. The genes upregulated in axons, along with those downregulated in cortex, were dominated by two functions: mitochondrial respiration and translation. Axons have high metabolic needs and abundant mitochondria, so it is unsurprising that enrichment of mitochondrial transcripts in axons has been reported by a number of studies (*Willis et al., 2007*; *Taylor et al., 2009*; *Gumy et al., 2011*; *Shigeoka et al., 2016*). Overall, 24% of the transcripts upregulated in axons and 25% of those downregulated in cortex encoded mitochondrial proteins, most of which were involved in either respiration or translation (*Figure 4d*, *Table 1*). A few mitochondrial genes were downregulated in axons, however, including some involved in regulation of mitochondrial fusion and

**Table 1.** Examples of genes found in auditory cortical axons during memory consolidation by function and effect of learning. Genes in bold type were changed in the opposite direction in the cortex.

| Type | Upregulated in axons | Downregulated in axons | Enriched in axons, not regulated |
|---|---|---|---|
| Mitochondrial respiration | **Atp5**(d,e,**g1,g2,g3**,h,i,j,**j2,5l,5o**), Atp6v(if,0b, **1g1**), Cox(**4l1**,5a,5b,6a1,6a2,**6c,7a2,7a2l,7b, 7 c**,8a,17), Dlst,Mdh1,**Mpc(1,2**), Ndufa(2,4,**5,6,7**,11,12,b1), **Ndufb**(2,3,4,5,**6,7**,8,**9,10,11**), **Ndufc2,Ndufs**(4,**5,6**,8), **Ndufv**(2,3),Suclg1, Uqcc2, Uqcr(10,**11,b**,c2,**fs1,h,q**) | Aco2,**Atp5**(a1,**b**),Fh,**Got2,Idh(2**,3b), **Ndufa10,Ndufs**(1,2,3),**Ndufv1, Ogdhl**, Pc,Pck2,**Pdh**(a1,b),**Sucla2** | Cox20,Me3,Uqcrc2 |
| Mitochondrial translation | **Mrp63,Mrpl**(11,12,13,16, **18,20**,23,27,**34**,35,**41**,44, 51,52,**54,55**),Mrps(7,11, 12,**14,15**,16,18b, 18 c, 21,23, **25,26**,28,**33**,34,36) | Mrpl(19,37),Mtif2,Tufm | Mrpl16,Mrps9 |
| Mitochondrial, other | **Fis1,Minos1, Timm(8b,10,13**) | Cluh,Immt,Mfn1,**Pink1**,Opa1 | Aldh2,Oxa1l,Sdhd |
| Ribosomal proteins | **Rpl**(3,4,5,6,7,8,9,**10**,10a,11,12,13,13a,14,15, 17,18,18a,**19,21,22l1**, 23,23a,**24,26,27,27a,28**, 29,**30,31,32,34,35,35a**, 36,36a,37,**38,39,p1**), Rps(3,**3a**,4x,5,**7,8**,10,11, 12,**13,14**,15,**15a**,16, 17, 18,21,23,**24,25**,26, 27, **27a,28**,29,a) | **Rps2** | |
| Translation apparatus/regulation | **Eef1**(a2,**b2**,d,e1),**Eif1b, Eif2s2**,Eif3g,Eif5b, **Erp29, Fkbp(2**,3),Hspa5,Naca, **Pfdn**(1,**2,5,6**), Sil1, **Srp(9**,14,**19**) | Apc,**Cyfip(1,2**),**Denr,Eef2**,Eif2b5, **Eif3**(a,**d**),Eif3l,Eif4a1,Eif6, **Mtor**,Rps6kb2,**Tsc2** | Rheb,Rps6ka2 |
| Spliceosome | Gemin7,**Hnrnp**(a1,**a2/b1**, a3,d,l,r,ul2),**Lsm(3,4,5**,7,8), Nono,**Sf3b(2,6**),Sfpq, Smndc1,**Snrnp27, Snrp(b2,c,d2,e,f,g**),Ssrf4 | Snrp200 | |
| Golgi apparatus | **Napg,Tmed9**,Trappc(3,5) | Copa,**Coro(1 c,7**),Gbf1, Gorasp1,**Trappc (9,10**,11) | Copg2 |
| Transcription | Brd(4,**7**),**Btf3**,Cited2, Ddit3,Dek,Dnajc2, **Drap1,Gtf2h5**,Hmgb1,Id4, **Lmo4**,Morf4l1,**Ncor(1,2**), Polr2(e,f,**g,j,k**),**Sub1**,Taf10 | Apbb1,Ahctf1,Baz2b, Cnot8,**Gtf3c(1**,3), Mta1, **Nsmf**,Polr2b | Baz1a,Hes6 |
| Proteasome/ubiquitination | Psm(a7,d4,d7,d12,g4), **Psmg4**, Ube2(k,v2) | Elp2,**Psm**(a1,**a4**,a5, b3,**b4,c1,c5**,d1,d2), **Ube**(3a,3b,**4b**),**Ubr4** | Psma6,Smurf1 |
| Active zone/synaptic vesicle cycle | Ap2s1,Bloc1s4,**Calm(1,2**), **Clta**,Gabbr1,Gng13, Hspa8,**Lin7b, Marcks**, Nos1ip,**Nrgn,Pfn**(1,3), **S100b,Stmn2**,Syt1,Unc13a | **Ap**(2a1,**2m1,3d1**), **Atp6v0a1,Brsk1,Bsn**, Btbd9,**Camk2a,Camkv**, Dnm1,Gna(12, b2,l1), **Gsn**,Nos1ap,Rab3a,**Scrib, Sptan1, Sptbn(1,2**), **Stxbp1**,Synj1,Vdac (1,2,3) | Nos1,Pcdh17,Prkcg |
| Cytoskeleton/axonal transport | **Bloc1s1,Dynll(1,2**), **Dynlrb1,Klc1, Sod1** | Bicd2,**Clip1,Dctn1,Dync (1h1**,2h1),Hap1,Htt, **Kif(3a**,5a,**5b,5c**,c3,ap3), **Myo**(1b,**1d, 5a**,5b,**9a**,9b,**16,18a**), **Myh(10,14**),Nefh, **Nefl**,Nefm, Tuba(1b,**4a**),Tubb(2b,**3,4a**, 4b,5) | Llgl1,Myh11,Myo10, Tubb2a,Tubg1 |
| Myelin sheath | | Ank3,Cnp,Cntnap1, **Mbp,Sptnb4** | Myrf |

*Table 1 continued on next page*

*Table 1 continued*

| Type | Upregulated in axons | Downregulated in axons | Enriched in axons, not regulated |
|---|---|---|---|
| Postsynaptic | | Dbn1,Ddn,Dlgap(1,3,4), Mink1,**Ppp1r9** (a,b), **Shank**(1,2,3) | |
| Other axonal/signaling | Akap5,Akip1,Eno1, **Gap43**,Mapt,**Olfm1**, Park7, Sumo2,Tmsb4x | **Akap**(2,6,8 l,**11**,13),Aldoc, **Arhgap**(21, 39),**Arhgef(2**,11), Dpysl2,**Fez1**,**Kalrn**, Rab(2b,3b,3c, 5 c,6b), Rock2,Vim | Arhgap26, Arhgef(12,18,28) |

localization, such as *Mfn1* and *Opa1*. The opposite pattern was reported in the transcriptome of cultured cortical neurons 2 days after injury: *Mfn1* was upregulated while transcripts related to respiration were downregulated (*Taylor et al., 2009*). If similar regulation occurs in the two paradigms, these results are consistent with translation of dormant axonal mRNAs in response to activity, leading to their upregulation in the translatome and subsequent depletion from the transcriptome.

Genes coding for translation-related functions, from mRNA splicing to protein folding, were also largely upregulated in axons and downregulated in cortex. Of 68 axonal transcripts encoding ribosomal proteins, 67 were upregulated after learning and 37 of these were downregulated in the cortex. The axonal translatome contained spliceosome components, nearly all of which were upregulated. Genes for initiation and elongation factors were mostly upregulated, although some were downregulated. Intriguingly, a number of genes encoding transcription factors were regulated in axons. Transcription factors are translated locally in growth cones and transported retrogradely to the nucleus (see *Ji and Jaffrey, 2014* for review), so this could be a case of developmental mechanisms supporting learning in the adult.

A number of transcripts encoding Golgi and rough ER proteins were present in the axonal translatome, although neither of these structures are seen in adult forebrain axons by EM. Similar observations have been reported in axons of cultured neurons, which carry out Golgi and rough ER functions in the absence of classical structures (*Willis, 2005*; *Merianda et al., 2009*; *González et al., 2016*). Rough ER proteins tended to be upregulated, whereas Golgi proteins were both upregulated and downregulated. Several upstream regulators of translation were downregulated in axons, including *Apc*, *Cyfip1*, *Mtor*, and *Tsc2*. Because mTOR complex 1 (mTORC1) activates translation of ribosomal proteins and translation factors (*Hsieh et al., 2012*; *Thoreen et al., 2012*; *Terenzio et al., 2018*), one possibility is that *Mtor* mRNA was depleted from axons in an initial wave of learning-associated translation, leading to upregulated translation of downstream targets at the time the tissue was collected. Consistent with this, IPA analysis indicated activation of mTOR in the axons (*Figure 4e*).

Mitochondrial and ribosomal genes made up half of the most highly expressed genes (*Figure 3—figure supplement 3c*), which could account for the high average expression level of upregulated axonal genes (*Figure 4*). However, removing these genes did not substantially lower the mean expression levels (*Figure 4—figure supplement 1d*), indicating that high expression is a feature of upregulated genes independent of function.

Genes downregulated in axons encoded more diverse types of proteins than upregulated genes. These included cytoskeletal components and molecular motors, including tubulins, myosins, dyneins, kinesins, and neurofilaments (*Figure 4d*, *Table 1*). Genes encoding synaptic proteins, including synaptic vesicle cycle, active zone, and postsynaptic density proteins, were downregulated, as were signaling molecules and components of the ubiquitin/proteasome pathway and myelin sheath. We used DAVID to examine the 25% of genes in our dataset that were not associated with learning to determine if there were any functions specific to these genes, but found only one term, 'mitochondrion,' enriched in axonal genes, and terms relating to the somatodendritic compartment enriched in the cortex (*Figure 3—figure supplement 4a*).

We compared the learning-associated genes to published translatomes of in vivo RGC axons (*Shigeoka et al., 2016*) and transcriptomes of cultured DRG and cortical axons (*Willis et al., 2007*; *Taylor et al., 2009*; *Gumy et al., 2011*), and found that genes that overlapped with only the

RGC axon translatome were twice as likely to be downregulated as upregulated; in contrast, the converse was true of genes in the cultured axon transcriptomes (*Figure 4f*). Regulated genes generally had more overlap with datasets from less mature axons, suggesting similar regulation of axonal translation during learning and development (*Figure 4—figure supplement 2b*). Upregulated genes were much more likely to overlap with genes downregulated rather than upregulated in response to injury (*Taylor et al., 2009*), consistent with similar translation patterns leading to depletion from the transcriptome.

## Transcript-level correspondence of axonal and cortical mRNA

Because alternative splicing could differ between the axons and cortex, we used Cufflinks software to compare expression at the transcript level. This analysis identified three genes that were not regulated at the gene level, but had one transcript upregulated (*Gng2*) or downregulated (*Snx27, Speg*) in axons while a second transcript was not (*Figure 4—figure supplement 3a*). Although multiple transcripts were identified for 133 (6%) of the 2185 differentially expressed genes, only one, *Gria2*, had one transcript significantly enriched in axons and another in cortex. Of the 656 genes that were associated with learning in both the axons and cortex, 54 had more than one transcript, and in 9nine cases, the same transcript was regulated in both (*Figure 4—figure supplement 3b–c*). To assess how learning-associated effects were distributed among transcripts in the two areas, we calculated a 'contribution score' for each transcript, indicating the fraction of the effect on its parent gene it represents. These scores were correlated between the axons and cortex (*Figure 4g*), indicating a high degree of coordinated regulation transcript level, similar to that seen at the gene level. Nevertheless, nine genes had transcripts whose axonal and cortical scores differed by >0.3, meaning that more than 30% of the learning effect was on different transcripts (*Figure 4—figure supplement 3b–c*).

## Imaging axonal mRNA

To verify axonal localization of mRNA in the amygdala in vivo, we used fluorescence in situ hybridization (FISH) combined with immunolabeling for axonal neurofilaments. We chose four transcripts that were abundant in control axons and significantly downregulated after learning: the Ras-related protein *Rab3a*, which regulates synaptic vesicle fusion, the N-myc downstream regulated gene *Ndrg4*, the Rab GDP dissociation inhibitor *Gdi1*, and *Ap2m1*, a subunit of the adaptor protein complex two which mediates synaptic vesicle endocytosis. We chose downregulated transcripts on the theory that these may represent constitutively translated genes in the control condition and would thus be less susceptible to varying translation levels over the course of consolidation. Successful FISH labeling required target retrieval treatments, including protease digestion, which proved incompatible with immunolabeling of cytoplasmic GFP in TE3 axons. The monoclonal antibody cocktail SMI312, which recognizes heavily phosphorylated axonal neurofilaments, was used to identify axons. Rats were given control training and brains were collected at the same time point as in the TRAP experiments. All four mRNA probes, but not the negative control probe, showed punctate labeling in the LA neuropil, with some puncta colocalized with axonal neurofilaments (*Figure 5*, *Figure 5—figure supplement 1*). Because the z-resolution of confocal microscopy may not be sufficient to unambiguously confirm colocalization, we repeated the labeling on 100 nm resin-embedded sections. The commercial FISH system we initially used did not work on these sections, so we used a traditional FISH protocol with an oligo(dT) probe. This probe revealed the expected pattern of poly-A RNA concentrated in cell bodies as well as both diffuse and punctate labeling throughout the neuropil, some of which colocalized with SMI312 (*Figure 6a*, *Figure 6—figure supplement 1*). Comparison of co-localization of the two labels revealed that a substantial amount of SMI312 label in the neuropil colocalized with oligo(dT) label (*Figure 6b*), while much less oligo(dT) was colocalized with SMI312. These observations further confirm the presence of mRNA in axons, and are consistent with the expectation that much of the oligo(dT) in the neuropil is in dendrites and glial processes.

## Results and discussion

Our results demonstrate that local translation occurs in axons of the adult forebrain in vivo, and that regulation of the axonal translatome within a memory circuit is associated with learning. This supports a growing body of evidence that mature axons are capable of local translation, contrary to

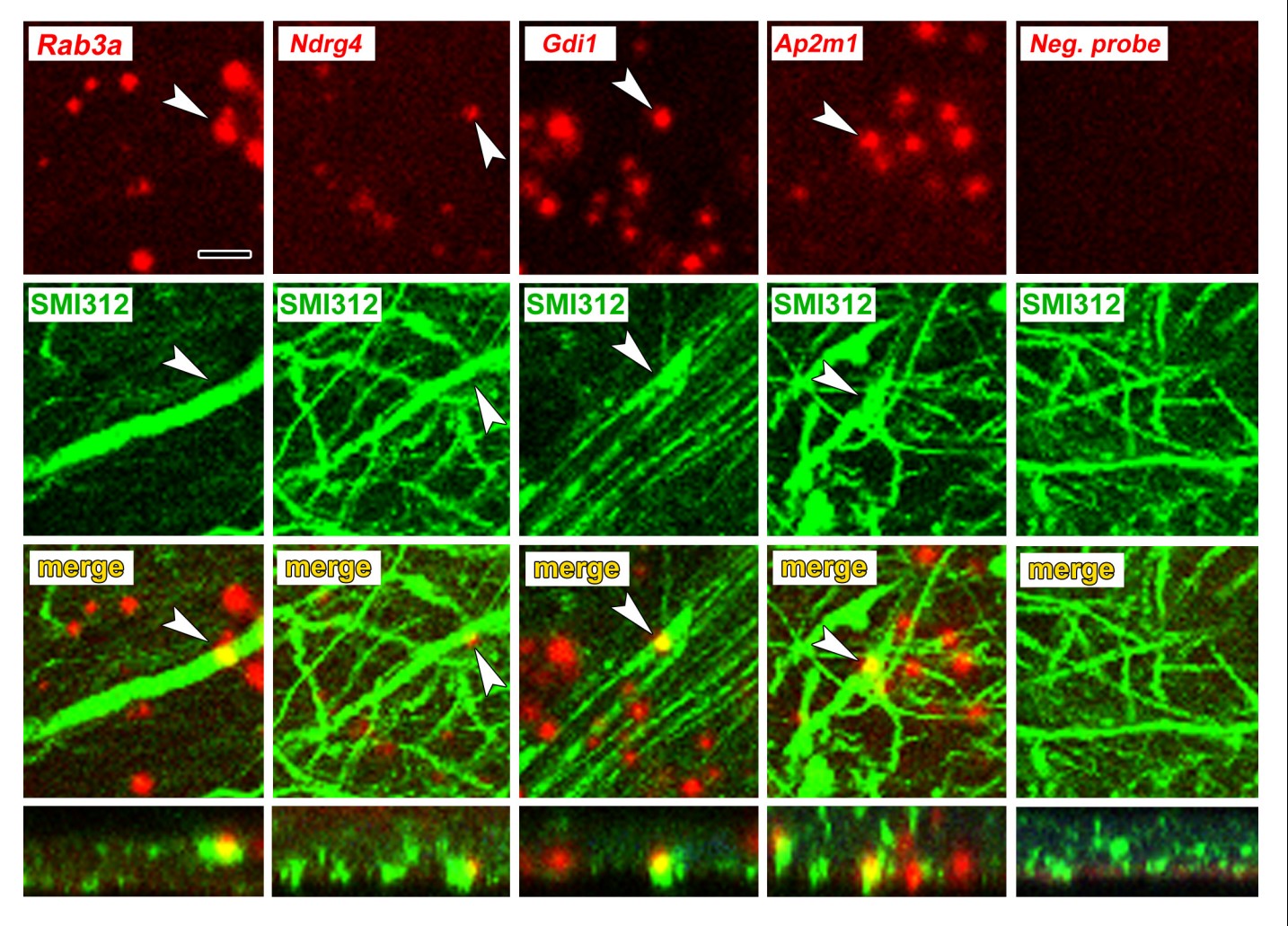

**Figure 5.** Axonal localization of mRNAs in vivo. First row: FISH showing localization of four mRNAs, but not a control probe, in amygdala neuropil. Second and third rows: Immunolabeling with the pan-axonal neurofilament antibody SMI312 shows overlap with mRNA probes. Bottom row: XZ orthogonal view of merged images. Scale = 5 µm.

The online version of this article includes the following figure supplement(s) for figure 5:

**Figure supplement 1.** Maximum intensity projections through 3 µm (10 confocal images with a 0.3 µm z-step size) of lateral amygdala showing FISH labeling and immunolabeling for neurofilaments.

traditional assumptions, and suggests that gene expression is more extensively decentralized than previously thought. A striking and unexpected feature of our data was the extent of opposing changes in the cortex and axons, suggesting highly coordinated regulation between the two compartments. In dendrites, mRNA transport is activity-regulated, with different trafficking mechanisms exist for different mRNAs (*Sutton and Schuman, 2006*; *Donnelly et al., 2010*; *Buxbaum et al., 2015*), and the axonal transcriptome could be similarly regulated. Neurotrophic factors have been shown to induce transport of existing mRNAs from the soma into the axons of cultured DRG neurons, and this is selective for transcripts encoding cytoskeletal proteins (*Willis, 2005*). The redistribution of transcripts from the soma to the axons could likewise be due to transport associated with learning. A large range of velocities has been reported for mRNA transport in neural processes (*Buxbaum et al., 2015*), and it is unknown whether mRNA travels from cortical cells to their distal projection fields in vivo in the timeframe of our experiment.

Because we analyzed ribosome-bound mRNAs, not the total mRNA in cortical cells, our data reflect not only mRNA localization but translation regulation as well. Downregulated transcripts may

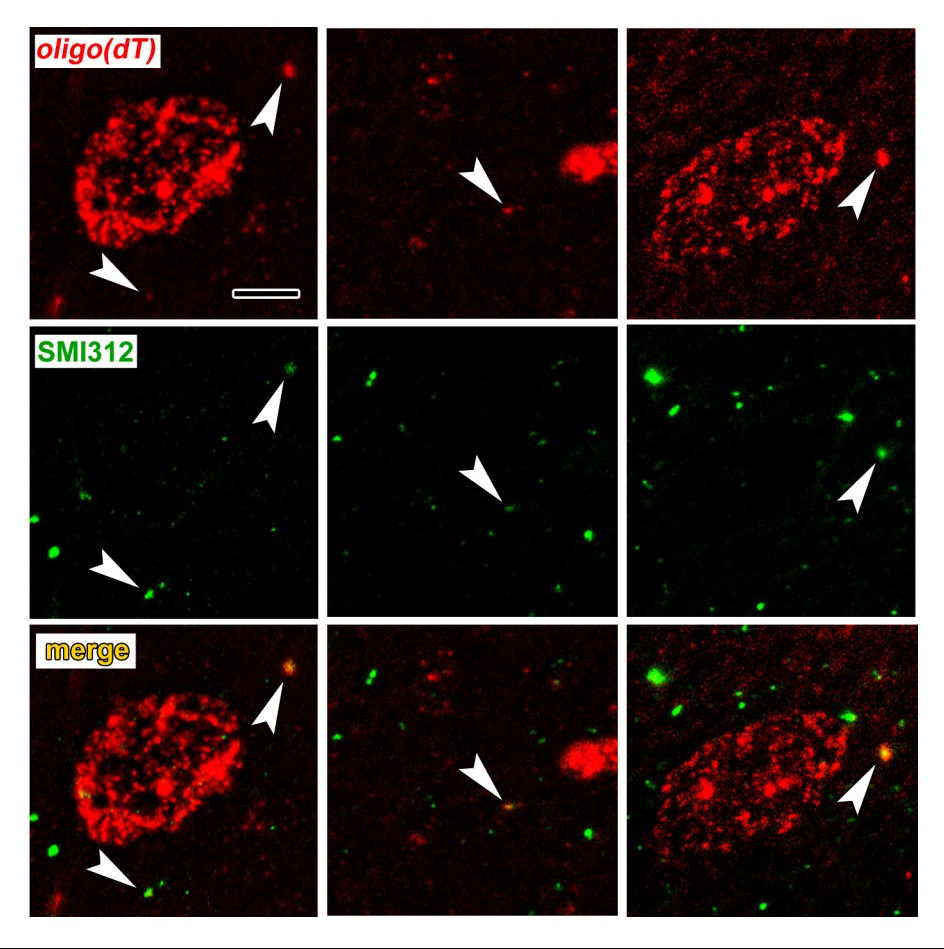

**Figure 6.** Colocalization of poly-A RNA with axonal neurofilaments in 100 nm resin-embedded amygdala sections. (a) Widefield images show overlap between and oligo(dT) probe and the pan-axonal neurofilament antibody SMI312 in the neuropil. Scale = 5 µm. (b–c) Mander's colocalization coefficients show a greater fraction of SMI312 signal colocalized with oligo(dT) in the neuropil versus the soma (b), but no difference oligo(dT) colocalized with SMI312 (c). * $F_{(1,32)}$ = 24.34, p=0.00002, $\eta^2$ = 0.43.

The online version of this article includes the following figure supplement(s) for figure 6:

**Figure supplement 1.** FISH labeling with an oligo(dT) probe combined with immunolabeling for neurofilaments on 100 nm amygdala sections.

reflect termination and subsequent degradation, whereas upregulated transcripts presumably represent new initiation, with or without new transcription. After initiation, ribosomes can be stalled on mRNAs, which are subject to regulated transport and reactivation (*Richter and Coller, 2015*). The TRAP method captures all ribosome-bound mRNA and cannot differentiate stalled ribosomes from those in an active elongation process at the moment of tissue harvest. Some of the mRNAs in our dataset are likely bound to stalled ribosomes, either because the transcripts are undergoing transport or are anchored in a dormant state awaiting reactivation. In addition, mRNAs can be transported and stored in a dormant state prior to initiation (*Buxbaum et al., 2015*). Rather than being newly trafficked from the soma, transcripts upregulated in the axons could result from unmasking of preexisting axonal mRNAs, and concomitant depletion from the cortex does not preclude upregulation of new, masked transcripts. Transcripts downregulated in the axons could reflect accelerated elongation in response to learning, or activation of stalled ribosomes, potentially with initiation and subsequent stalling of transcripts in the cortex to replenish the axonal supply. Finally, it should be noted that with the TRAP method, it is possible that the mRNAs that are isolated could be

extraribosomal. Thus, additional work with multiple approaches will be required to elucidate the full extent of the axonal transcriptome and the dynamics of its translation.

Our cortical samples contained intrinsic and corticocortical axons, and it is therefore possible that some of our data derive from asynchronous changes in proximal versus distal axons, potentially due to more rapid trafficking of mRNA from the soma or differential regulation in the proximal axons. We found an assortment of translation initiation factors and genes coding for them, along with spliceosome components, in axons, making it likely that at least some axonal translation is locally initiated. The presence of genes associated with structures surrounding axons, such as myelin basic protein (*Mbp*), spinophilin (*Ppp1r9b*), dendrin (*Ddn*), and the shank proteins (*Shank1, 2, and 3*), could reflect previously unknown axonal functions of these proteins, as perhaps evidenced by the presence of *Mbp* mRNA in unmyelinated cultured axons (*Gumy et al., 2011*). Alternatively, this could result either from trans-endocytosis between dendritic spines and axonal boutons (*Spacek and Harris, 2004*) or exosomal transfer between myelin and the axon shaft (*Giuditta et al., 2008*; *Twiss and Fainzilber, 2009*). Translation regulation in axons is likely to be extensively regulated through multiple mechanisms, the details of which are yet to be fully discovered.

A potential source of bias in our data is the use of RPL10a to capture ribosome-bound mRNA. There is emerging evidence that some ribosomal proteins preferentially translate particular subsets of mRNAs (*Xue and Barna, 2012*), including RPL10a, which has been found to have a bias for genes associated with the extracellular matrix and development in mouse embryonic stem cells (*Shi et al., 2017*). If similar ribosome selectivity occurs in the adult brain, our dataset may not reflect the full diversity of the axonal translatome. Our viral TRAP approach relies on overexpression of the modified RPL10a protein, so it is possible that the complement of ribosomal proteins is altered with more RPL10a in axons than normal. Translation is subject to a very high degree of regulation, most of which occurs at the initiation step, and the rate-limiting factor is eIF4E, which recruits ribosomes to mRNA (*Groppo and Richter, 2009*; *Sonenberg and Hinnebusch, 2009*). Therefore, excess ribosomal proteins would not be expected to alter translation dynamics. Consistent with this prediction, the original report of the TRAP technique found no functional differences between overexpressed RPL10a-eYFP and endogenous RPL10a (*Heiman et al., 2008*).

The differences we observed between our trained and control groups can be attributed generally to learning, but our data do not address the potentially different effects of various forms of experience-dependent plasticity on the axonal translatome. The Pavlovian conditioning protocol that we used produces a specific type of associative learning, in which an excitatory relationship between the tone and the shock is established. A commonly used control for this procedure is unpaired training, which explicitly separates the tones and shocks so that the tones never predict the shocks. Unpaired training is itself an associative learning paradigm, however: it produces both an excitatory association between the training context and the shock and an inhibitory association between the tone and shock, with corresponding changes in LA synapses (*Rogan et al., 2005*; *Ostroff et al., 2010*). Because there were no reference datasets to serve as benchmarks for an axonal translatome of an adult animal under quiescent conditions, we chose not to compare associative learning paradigms, but instead to use a control group that did not undergo learning. There is no way to entirely avoid learning when novel stimuli are presented; in the absence of shocks, exposure to tones induces remapping of the receptive fields of auditory cortical neurons that reduces responses to the habituated tone (*Condon and Weinberger, 1991*), and also produces latent inhibition, a form of learning that produces a null association to the stimulus and impairs subsequent associations (*Weiner and Feldon, 1997*). Thus, exposing the control group to tones on the training day, even without shocks, would have resulted in learning. Although habituation to the tone ahead of time would mitigate this, it would result in altered auditory circuits and require equal habituation of the trained group, which would interfere with learning. Our data therefore do not solely represent effects specific to the excitatory association, but likely include effects that are broadly induced by different forms of learning. More targeted experiments will be needed in the future to isolate and compare changes in the axonal translatome that are specific to excitatory versus inhibitory associations, and non-associative learning.

The spatiotemporal uncoupling of translation from transcription has unique implications in the brain, which is itself functionally compartmentalized. The increasing use of gene expression to catalog cells and brain areas, along with genetic targeting of brain circuits, will need to be reexamined if axonal translation is widespread in the adult brain. The idea that translation can be spatially

regulated has gradually gained acceptance in a number of contexts, but these continue to be considered exceptional circumstances. Our results counter the longstanding assumption that axonal translation does not occur in the adult brain, and the number and variety of transcripts we identified suggests that spatial regulation could be a fundamental component of translation.

# Materials and methods

### Key resources table

| Reagent type | Designation | Source | Identifiers | Additional information |
|---|---|---|---|---|
| Antibody | Rabbit polyclonal anti-eIF4E | Bethyl Labs | Cat# A301-154A | 1:500 (EM-IHC) |
| Antibody | Mouse polyclonal anti-eIF4G1 | Abnova | H00001981-A01 | 1:500 (EM-IHC) |
| Antibody | Mouse monoclonal anti-eIF2α | Cell Signaling | L57A5 | 1:500 (EM-IHC) |
| Antibody | mouse monoclonal anti-GFP | Invitrogen | A11120 | 1:1000 (EM-IHC) |
| Antibody | mouse monoclonal anti-neurofilament cocktail | BioLegend | SMI312 | 1:500 (IHC); 1:250 (IHC) |
| Antibody | mouse monoclonal anti-GFP | PMID: 24810037 | RRID: AB_2716736 | .29µg/µl |
| Antibody | mouse monoclonal anti-GFP | PMID: 24810037 | RRID: AB_2716737 | .29µg/µl |
| Recombinant DNA reagent | pAAV-CMV-eYFP-L10a | PMID: 24904046 | | Dr. Thomas Launey (RIKEN Brain Science Institute) |
| Recombinant DNA reagent | VSVG.HIV.SIN.cPPT.CMV.eYFP-L10a | this paper (Materials and methods) | | |
| Software, algorithm | Fiji | PMID: 22743772 | RRID: SCR_002285 | |

### Subjects, surgery, and behavior

All animal procedures were approved by the Animal Care and Use Committees of New York University and the University of Connecticut. Subjects were adult male Sprague-Dawley rats weighing ~300 g, housed singly on a 12 hr light/dark cycle with ad libitum food and water. All procedures were performed during the rats' light cycle. For virus injections, rats were anesthetized with ketamine/xylazine and given bilateral stereotaxic injections of either 0.2 µl AAV-CMV.eYFP-L10a or 1 µl lenti-CMV.eYFP-L10a or lenti-CMV.eYFP (Emory Neuroscience Viral Vector Core) into TE3 (AP 3.8, ML 6.8, DV 3.7 mm from interaural center) using a Hamilton syringe. Animals were given at least two weeks to recover from surgery before experiments began.

Behavioral training took place in a soundproof, lit 28.5 × 26×28.5 cm chamber (Coulbourn Instruments). Auditory tones (30 s, 5 kHz, 80 dB) were delivered through a speaker inside the chamber, and footshocks (0.7mA, 1 s) were delivered through a grid floor. Rats were habituated to the conditioning chamber for 30 min for 2 days prior to training. The training protocol consisted of five tones co-terminating with foot shocks delivered over 32.5 min with a variable interval between tone-shock pairings.

### Immunolabeling and electron microscopy

Rats were deeply anesthetized with chloral hydrate (1.5 mg/kg) and perfused transcardially with 500 ml of mixed aldehydes at pH 7.4 at a rate of 75 ml/min with a peristaltic pump. For eYFP immunolabeling, two lentivirus-injected and two uninjected rats were perfused with 0.25% glutaraldehyde/4% paraformaldehyde/4 mM $MgCl_2$/2 mM $CaCl_2$ in 0.1M PIPES buffer. For eIF4E and eIF4G labeling six rats were perfused with 0.5% glutaraldehyde/4% paraformaldehyde/4 mM $MgCl_2$/2 mM $CaCl_2$ in

0.1M PIPES buffer and alternate sections were used for each antibody. For eIF2α six rats were perfused with 0.25% glutaraldehyde/4% paraformaldehyde in 0.1M phosphate buffer. For ribosomal protein S6 labeling, three rats were perfused with 0.1% glutaraldehyde/4% paraformaldehyde in 0.1M phosphate buffer. Aldehydes and PIPES buffer were obtained from Electron Microscopy Sciences, phosphate buffer and salts were from Sigma-Aldrich. Brains were removed and immersed in the perfusion fixative for one hour before rinsing in buffered saline (0.01M fixation buffer with 154 mM NaCl) and sectioning at 40 µm on a vibrating slicer. Sections were blocked for 15 min in 0.1% sodium borohydride, rinsed in buffer, and blocked in 1% bovine serum albumin (BSA; Jackson Labs) before overnight incubation in primary antibody in 1% BSA at room temperature. Sections were rinsed, incubated in 1:200 biotinylated goat anti-rabbit or goat anti-mouse (Vector Labs) in 1% BSA for 30 min, rinsed, incubated in avidin/biotin complex peroxidase reagent (Vector Labs Vectastain Elite ABC PK-6100) for 30 min, then reacted 5 min with 1 mM 3,3 diaminobenzidine in 0.0015% $H_2O_2$.

All sections from the brains injected with LV-CMV-eYFP-L10a were examined to confirm that there were no infected cell bodies outside of the TE3 injection site. The area around the LA was dissected out of the immunolabeled sections for electron microscopy. Tissue was processed for electron microscopy as previously described (*Ostroff et al., 2010*). Briefly, tissue was postfixed in reduced osmium (1% osmium tetroxide/1.5% potassium ferrocyanide) followed by 1% osmium tetroxide, dehydrated in a graded series of ethanol with 1.5% uranyl acetate, infiltrated with LX-112 resin in acetone, embedded in resin, and cured at 60° for 48 hr. 45 nm sections were cut on an ultramicrotome (Leica) and imaged on a JEOL 1200EX-II electron microscope at 25,000X on an AMT digital camera. Images were cropped and contrast adjusted using Photoshop (Adobe).

For quantification of eIF4E immunolabel, serial 45 nm sections (average 97+ /- 5) were imaged from each of the six samples. A 4 × 4 µm square was defined in the middle of the central section of each series, and every profile within the square was followed through serial sections to determine its identity and whether it contained label within the series. If a profile could not be definitively identified as an axon, dendrite, spine, or glial process within the series, it was classified as unidentified.

## Antibodies

Antibody sources and dilutions for immunohistochemistry were as follows: anti-eIF4E rabbit polyclonal (Bethyl Labs A301-154A, lot# A301-154A-1) 1:500, anti-eIF4G1 mouse polyclonal (Abnova H00001981-A01, lot# 08213-2A9) 1:500, anti-eIF2α mouse monoclonal (Cell Signaling L57A5, lot# 3) 1:500, anti-GFP mouse monoclonal (Invitrogen A11120, clone# 3E6) 1:1000, and anti-neurofilament (highly phosphorylated medium and heavy) mouse monoclonal cocktail (BioLegend SMI312 Lot# B263754) 1:1000 for RNAscope experiments and 1:250 for labeling 100 nm resin sections. To confirm antigen recognition by the polyclonals to eIF4E and eIF4G, the primary antibodies were preadsorbed before use with a 10-fold excess of the immunizing peptide obtained from the antibody supplier, which reduced the density of labeled structures by 97–98%. To control for specificity of the GFP antibodies, tissue from animals without viral injections was run in parallel and did not result in labeled structures. For immunoprecipitation of eYFP-L10a, two mouse monoclonal anti-GFP antibodies (HtzGFP-19F7 lot# 1/BXC_4789/0513 and HtzGFP-19C8 lot# 1/BXC_4788/0513; available from the Memorial Sloan-Kettering Cancer Center Monoclonal Antibody Core Facility, New York, NY) were used as described below. SMI312 is a cocktail of affinity-purified mouse monoclonal antibodies that recognize highly phosphorylated medium and heavy neurofilament polypeptides.

## Cloning and virus packaging

pAAV-CMV-eYFP-L10a was a generous gift from Dr. Thomas Launey (RIKEN *Kratz et al., 2014*). YFP-L10a was excised from pAAV-CMV-eYFP-L10a using Nhe I and Xho I. The ~1.4 kb band was gel purified (QiaQuick Gel Extraction Kit, Qiagen, Hilden, Germany). pLV-eGFP (purchased from Adgene) was digested with Xba I and Sal I, and the ~6.7 kb band was gel purified. The eYFP-L10a and pLV backbone were then ligated according to the manufacturer's protocol (T4 DNA ligase, ThermoFisher Scientific, Springfield Township, NJ). Virus (VSVG.HIV.SIN.cPPT.CMV.eYFP-L10a) was packaged by The University of Pennsylvania Vector Core. Viral titer was 2.29e09 GC (genome copies)/mL.

## Immunoprecipitation and RNA isolation

Exactly 90 min after the completion of behavioral training, rats (n = 10 per group) were deeply anesthetized with chloral hydrate (1.5 mg/kg) and perfused transcardially with 20 ml ice cold oxygenated artificial cerebrospinal fluid (ACSF) consisting of 125 mM NaCl, 3.3 mM KCl, 1.2 mM $NaH_2PO_4$, 25 mM $NaHCO_3$, 0.5 mM $CaCl_2$, 7 mM $MgSO_4$, and 15 mM glucose with 50 µM cycloheximide. Brains were quickly removed, blocked coronally around the amygdala and auditory cortex, and the two hemispheres separated and incubated in the perfusion solution for 4–5 min. Each hemisphere was then bisected along the rhinal fissure. The cortex of the dorsal half was peeled away from the underlying hippocampus and the area containing TE3 was dissected out. A smaller block containing the amygdala was dissected from the ventral half by peeling away the ventral hippocampus, trimming off the cortex lateral to the external capsule and trimming away the hypothalamus and medial portion of the striatum. The TE3 and amygdala blocks were quickly frozen in liquid nitrogen and stored at −80℃. Control and trained animals were run in parallel and tissue was collected in the middle of the animals' light cycle.

The polysome purification and RNA extraction were performed according to published protocols (*Heiman et al., 2008*; *Kratz et al., 2014*). TE3 or amygdala tissues from five animals were pooled (resulting in two biological replicates per group for sequencing), as pilot experiments found that this yielded sufficient mRNA. Samples were homogenized in 2 ml of ice-cold polysome extraction buffer [10 mM HEPES, 150 mM KCl, 5mMMgCl2, 0.5 mM DTT, one minitablet Complete-EDTA free Protease Inhibitor Cocktail (Roche), 100 µl RNasin Ribonuclease Inhibitor (Promega) and 100 µl SUPERase In RNase inhibitor (Ambion), 100 µg/ml cycloheximide] in douncer homogenizer. Homogenates were centrifuged for 10 min at 2000 x g at 4℃. The supernatants were clarified by adding 1% IGEPAL CA-630 (SigmaAldrich) and 30 mM DHPC (Avanti Polar Lipids) and incubated for 5 min on ice. The clarified lysates were centrifuged for 15 min at 20,000 x g at 4℃ to pellet unsolubilized material, and 100 µl of the supernatant fluid was collected for isolation of the tissue transcriptome. The remainder was added to the conjugated beads/antibodies (200 µl) and incubated at 4C overnight with gentle agitation. The following day, the beads were collected with magnets for 1 min on ice, then washed in 1 mL 0,35M KCl washing buffer (20 mM HEPES, 350 mM KCl, 5mMMgCl₂, 0.5 mM DTT, 1% IGEPAL CA-630, 100 µl RNasin Ribonuclease Inhibitor and 100 µl SUPERase In RNase inhibitor, 100 µg/ml cycloheximide) and collected with magnets.

The conjugated beads/antibodies were freshly prepared before the homogenization on the day of the experiment by incubating 300 µl of Dynabeads MyOne Streptavidin T1 (ThermoFisher Scientific) with 120 µl of 1 µg/µl Biotinylated Protein L (ThermoFisher Scientific) for 35 min at room temperature with gentle rotation. Then, the conjugated protein L-beads were washed with 1XPBS and collected with magnets for three times. The conjugated protein L-beads were resuspended in 175 µl of 0.15M KCl IP wash buffer (20 mM HEPES, 150 mM KCl, 5mMMgCl₂, 0.5 mM DTT, 1% IGEPAL CA-630, 100 µl RNasin Ribonuclease Inhibitor and 100 µl SUPERase In RNase inhibitor, 100 µg/ml cycloheximide) and incubated for 1 hr at room temperature with 50 µg of each antibody. The beads were then washed 3 times with 0.15M KCl IP wash buffer and resuspended in the same buffer with 30 mM DHPC.

The RNA was extracted and purified with Stratagene Absolutely RNA Nanoprep Kit (Agilent Technologies, Santa Clara, CA) according to the manufacturer's instructions. All the buffers were provided with the kit except otherwise specified. Briefly, the beads were resuspended in Lysis Buffer with ß-mercaptoethanol, incubated for 10 min at room temperature. 80% Sulfolane (Sigma) was added to the samples and the samples were mixed for 5–10 s, then added to an RNA-binding nanospin cup and washed with a Low Salt Washing Buffer by centrifuge for 1 min at 12,000 x g at room temperature. DNA was digested by mixing the DNase Digestion Buffer and the samples for 15 min at 37C. Then, the samples were washed with High-Salt Washing Buffer, Low-Salt Washing Buffer and centrifuged for 1 min at 12,000 x g. Finally, the samples were eluted with Elution Buffer and centrifuge for 5 min at 12,000 x g at room temperature. The isolated RNA was stored at −80℃.

## Sequencing and differential gene expression (DGE) analysis

RNASeq libraries were made using the SMART-Seq v4 Ultra Low Input RNA Kit for Illumina Sequencing, with the Low Input Library Prep kit v2 (Clontech, Cat # 634890 and 634899, respectively), using 50–200 pg of total RNA. 16 cycles of PCR were used for the cDNA amplification step, and 5 PCR

cycles to amplify the library prep. Libraries were run on an Illumina HiSeq 2500 instrument, using a paired end 50 protocol; eight samples were pooled per lane of a high output paired end flow cell, using Illumina v4 chemistry.

Raw sequencing data were received in FASTQ format. Read mapping was performed using Tophat 2.0.9 against the rn6 rat reference genome. The resulting BAM alignment files were processed using the HTSeq 0.6.1 python framework and respective rn6 GTF gene annotation, obtained from the UCSC database. Subsequently the Bioconductor package DESeq2(3.2) was used to identify differentially expressed genes (DEG). This package provides statistics for determination of DEG using a model based on the negative binomial distribution. The resulting values were then adjusted using the Benjamini and Hochberg's method for controlling the false discovery rate (FDR). Genes with an adjusted p-value<0.05 were determined to be differentially expressed. For transcript-level analysis, the Cufflinks suite (version 2.2.1) was used. ANOVAs and post hoc Bonferroni tests were run using the STATISTICA software package (StatSoft). Raw sequencing data and analysis are available in the NCBI Gene Expression Omnibus (accession # GSE124592).

## Filtering of DGE results

To isolate the axonal translatome with as few false positives as possible, we employed a stringent filtering strategy to our DGE data. Twelve comparisons were run between the eight samples: the TRAPed mRNAs from the axons and cortex were compared to each other separately in each of the training conditions, and the conditions were compared to each other separately in the two brain areas. The same analysis was performed on the tissue transcriptome samples, and each of the four TRAPed samples was compared directly to its corresponding transcriptome. To assemble a list of axonal mRNAs, we began with the comparisons between the TRAPed samples, since this should account for much of the IP background. Because of potential background noise and variability between the individual samples preparations, we excluded genes from each TRAP comparison if the same effect was observed in the corresponding transcriptome comparison. In addition, genes enriched in a given comparison between TRAP samples were excluded if they were not also enriched versus the transcriptome. Although both of these steps likely result in many false negatives, particularly among transcripts that are highly abundant or ubiquitous in the tissue, we felt that excluding potential false positives was crucial given the novelty of our dataset.

## Gene ontology and ingenuity pathway analysis

Gene lists were submitted to the DAVID (*Huang et al., 2009*) Functional Annotation Chart tool and enrichment data from the GOTERM_BP_DIRECT (biological process), GOTERM_CC_DIRECT (cellular component), and GOTERM_MF_DIRECT (molecular function) gene ontology categories and KEGG_-PATHWAY (Kyoto Encyclopedia of Genes and Genomes) category were examined, using a Benjamini-Hochberg adjusted p-value cutoff of <0.05. For comparison of learning effects, all regulated genes in each area were submitted, regardless of any effect or enrichment in the other area.

For Ingenuity Pathway Analysis (Qiagen Bioinformatics), we submitted all genes differentially expressed (adjusted p-value<0.05) between the training groups in the axons and cortex, along with the corrected $\log_2$(fold change) calculated by DESeq2. We performed a Core Analysis with the reference data restricted to human, mouse and rat genes and nervous system tissue; otherwise the program's default settings were used.

## Fluorescence in situ hybridization on fixed sections

Adult male rats (n = 4) were given control training and perfused 90 min later with 4% paraformaldehyde in 0.1M phosphate buffer, pH 7.4. Brains were sectioned at 40 μm on a vibrating tissue slicer (Leica) and mounted on glass slides. RNA was detected using the RNAscope 2.5 HD RED kit (Advanced Cell Diagnostics, Inc) according to the manufacturer's instructions, with the exception that the incubation time for the fifth amplification step was doubled to increase the diameter of the puncta. Each section was labeled with one of five probes: *Rab3a*, *Ndrg4*, *Ap2m1*, *Gdi1*, or *DapB* (negative control). Sections were blocked overnight in 1% bovine serum albumin with 0.1% Triton-X in phosphate buffered saline, then incubated with primary antibody for 48 hr at 4° followed by 1:200 Alexa-488 goat anti-mouse for 1 hr at room temperature. Slides were stained with DAPI, mounted in Prolong Gold (Invitrogen), and imaged on a Leica TCS SP8 confocal microscope (Leica

Microsystems). Z stacks were collected using a 63 × 1.40 HC PL APO oil immersion lens and z step size of 0.3 microns. All sections were stained in parallel with the same batches of probes and antibody. Laser intensity and gain were constant for all images and brightness and contrast were not adjusted. Maximum intensity projections were created in ImageJ.

### Fluorescence in situ hybridization on resin-embedded sections

Adult male rats (n = 3) were perfused transcardially with 4% paraformaldehyde and 0.1% glutaraldehyde in 0.1M phosphate buffer, pH 7.4. Brains were sectioned at 100 µm on a vibrating slicer and sections containing the amygdala were dissected. Sections were dehydrated through ascending ethanol dilutions, infiltrated with LR White resin (Electron Microscopy Sciences), and cured at 60˚ for 48 hr. 100 nm sections were cut on an ultramicrotome (Leica) and mounted on gelatin-coated glass slides for labeling. Sections were hydrated in PBT (PBS-pH7.4 + 0.1% Tween20) at room temperature for 5 min (3X) followed by incubation with oligo(dT)-ATTO633 probe (Integrated DNA Technologies) in hybridization buffer (5X SSC + 1X-Denhardt's buffer (Sigma – D2532) + 5% Dextran Sulfate + 0.05M Phosphate buffere pH6.7 + 0.1% SDS) at 60C in a hydrated chamber. Post hybridization washes were done at 60C in 1X SSC + 50% deionized formamide/10 min; 2X SSC/5 min; 0.2X SSC/2 min. Sections were then washed in PBT for 5 min at room temperature followed by fixation in 4% paraformaldehyde for 10 min at room temperature. After washing the sections twice in PBT for 5 min at room temperature, they were blocked in PBT + 0.5% bovine serum for 5 min at room temperature. Sections were incubated in primary antibody followed by 1:200 Alexa-488 donkey anti-mouse, both for 30 min in the blocking solution. After blocking in 0.5% bovine serum albumin in phosphate buffered saline containing 0.1% Tween-20, sections were incubated in primary antibody followed by 1:200 Alexa-488 donkey anti-mouse, both for 30 min in the blocking solution. Slides were mounted in Prolong Gold and imaged on a Nikon Eclipse TiE microscope (Nikon Instruments) with a Photometrics Prime 95B CMOS camera (Teledyne Photometrics) using a 100 × 1.49 PL APO oil immersion lens. Colocalization was quantified using the Coloc2 plugin in Fiji. (*Schindelin et al., 2012*) Background was removed from the images by subtracting the mean intensity of the no-primary antibody control images from the SMI312 channel, and subtracting the mean intensity of images of the resin surrounding the tissue from the oligo(dT)-ATTO633 channel. ROIs were drawn to segment the neuropil and somata (n = 17 per region) and thresholded using the Costes method. (*Costes et al., 2004*) One-way ANOVAs were used to compare Manders colocalization coefficients (*Manders et al., 1993*) for each channel between the soma and neuropil.

## Acknowledgements

The L10a-YFP construct was a generous gift from Dr. Thomas Launey at the RIKEN Brain Science Institute, Tokyo, Japan. We are grateful to Yutong Zhang for expert technical assistance and to Drs. Joel Richter and Erin Schuman for their insightful comments on the manuscript. This work was supported by NIH grants MH119517, MH083583 and MH094965 to LO, NS087112 to ES, and NS034007, NS047384, and HD082013 to EK. We thank the Applied Bioinformatics Laboratories (ABL) at the NYU School of Medicine for providing bioinformatics support and helping with the analysis and interpretation of the data. This work has used computing resources at the NYU High-Performance Computing Facility (HPCF) and was supported in part by the Viral Vector Core of the Emory Neuroscience NINDS Core Facilities grant, P30NS055077. The Leica SP8 confocal used in this study was obtained with a grant from the NIH (S10OD016435) awarded to Akiko Nishiyama.

## Additional information

### Funding

| Funder | Grant reference number | Author |
| --- | --- | --- |
| National Institute of Neurological Disorders and Stroke | NS034007 | Eric Klann |
| Eunice Kennedy Shriver National Institute of Child Health and Human Development | HD082013 | Eric Klann |

| National Institute of Mental Health | MH083583 | Linnaea E Ostroff |
| National Institute of Neurological Disorders and Stroke | NS047384 | Eric Klann |
| National Institute of Mental Health | MH094965 | Linnaea E Ostroff |
| National Institute of Mental Health | MH119517 | Linnaea E Ostroff |
| National Institute of Neurological Disorders and Stroke | NS087112 | Emanuela Santini |

The funders had no role in study design, data collection and interpretation, or the decision to submit the work for publication.

## Author contributions

Linnaea E Ostroff, Conceptualization, Resources, Formal analysis, Supervision, Funding acquisition, Investigation, Visualization, Methodology, Project administration; Emanuela Santini, Conceptualization, Funding acquisition, Investigation, Methodology; Robert Sears, Zachary Deane, Rahul N Kanadia, Investigation, Methodology; Joseph E LeDoux, Conceptualization, Resources; Tenzin Lhakhang, Formal analysis; Aristotelis Tsirigos, Data curation, Formal analysis, Methodology; Adriana Heguy, Formal analysis, Investigation; Eric Klann, Conceptualization, Resources, Funding acquisition

## Author ORCIDs

Linnaea E Ostroff https://orcid.org/0000-0002-3348-342X
Eric Klann https://orcid.org/0000-0001-7379-6802

## Ethics

Animal experimentation: All animal procedures were performed in accordance with the guidelines in the National Institutes of Health Guide for the Care and Use of Laboratory Animals, and were approved by the Animal Care and Use Committees of New York University (protocol 01-1097) and the University of Connecticut (protocol A17-036).

## Decision letter and Author response

Decision letter https://doi.org/10.7554/eLife.51607.sa1
Author response https://doi.org/10.7554/eLife.51607.sa2

## Additional files

### Supplementary files

- Supplementary file 1. RNA quality control data.

- Supplementary file 2. Results of differential gene expression analysis and subsequent filtering.

- Supplementary file 3. Results of differential gene expression analysis and subsequent filtering, YFP samples.

- Supplementary file 4. Results of DAVID enrichment analyses of all axonal genes, cortex-only genes, and genes that were upregulated and downregulated in the axons and cortex.

- Supplementary file 5. Results of ANOVA and post hoc Bonferroni test comparing mean FPKM between experimental groups by training effect.

- Supplementary file 6. Results of IPA upstream regulator analysis of training effects in axons and cortex.

- Supplementary file 7. Results of IPA functional annotation analysis of training effects in axons and cortex.

- Supplementary file 8. Transcript-level FPKM values and results of differential expression analysis.

- Transparent reporting form

## Data availability

Sequencing data have been deposited in GEO under accession code GSE124592. All analyses are included in supporting files.

The following dataset was generated:

| Author(s) | Year | Dataset title | Dataset URL | Database and Identifier |
|---|---|---|---|---|
| Ostroff L, Klann E | 2018 | The translatome of adult cortical neurons is regulated by learning in vivo | https://www.ncbi.nlm.nih.gov/geo/query/acc.cgi?acc=GSE124592 | NCBI Gene Expression Omnibus, GSE124592 |

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
