## [Decision Letter]

**Acceptance summary:**

Local translation is important for memory consolidation by supplying new proteins to new synaptic connections to stabilize them. This work demonstrates that translation in adult forebrain dendrites is regulated by learning. Axons in the adult rat amygdala contain translation machinery, and the authors used translating ribosome affinity purification (TRAP) with RNASeq to identify mRNAs in cortical axons regulated during associative memory. Mitochondrial and translation-related genes were upregulated, whereas synaptic, cytoskeletal, and myelin-related genes were downregulated. Their results demonstrate that learning-regulated axonal translation occurs in the adult forebrain.

These are important observations for the neuroscience field, especially for neuronal-oriented cell biologists in the areas of learning and memory. It will also be of interest for RNA biologists interested in translational regulation because it opens the pathway to investigating the mechanisms by which translation is regulated in complex cells such as neurons.

**Decision letter after peer review:**

Thank you for submitting your article "The translatome of adult cortical axons is regulated by learning in vivo" for consideration by *eLife*. Your article has been reviewed by three peer reviewers, one of whom is a member of our Board of Reviewing Editors, and the evaluation has been overseen by Eve Marder as the Senior Editor. The following individuals involved in review of your submission have agreed to reveal their identity: Pablo E Castillo (Reviewer #2); Christine E Holt (Reviewer #3).

The reviewers have discussed the reviews with one another and the Reviewing Editor has drafted this decision to help you prepare a revised submission.

Summary (from reviewer 3):

This manuscript investigates the question of whether translation in the presynaptic compartment in the mammalian cortex is altered by learning. The authors provide immuno EM evidence that axons in the rat amygdala contain translation-associated machinery and use TRAP and RNASeq to characterise the ribosome-bound mRNAs in cortical axons projecting to the amygdala under different learning paradigms. They report that over 1200 mRNAs are regulated during the consolidation of associative memory. Mitochondrial and translation related genes, particularly, were upregulated whereas other genes such as synaptic, cytoskeletal and myelin-related, were downregulated. Also, of interest, opposite up/down gene regulation occurred in the cortex. The authors conclude that learning-regulated translation changes occur in axons of the forebrain and suggest that they may be widespread and important for learning.

Essential revisions:

While the reviewers find the work interesting and important, they feel that there should be more evidence that learning leads to local translation and subsequent presynaptic plasticity: that localization of mRNAs and their translation in axons can be related to memory consolidation. They suggest higher resolution imaging and use of more specific antibodies as possible approaches. In particular, reviewer 2 suggests an additional control and would like higher resolution evidence that the mRNAs are actually in axons. Additional suggestions include using translation indicators (e.g. antibodies for phosphorylated initiation factors). Reviewer 3 has suggestions concerning how the data was analyzed.

*Reviewer #1:*

In the manuscript submitted by Ostroff et al., the authors describe the identity of axonal translatome within the lateral amygdala following auditory fear condition. Intriguingly, the results show an increase in the synthesis of mitochondrial and translation machinery-related proteins after learning. Moreover, the selective translation of such genes was most similar to the translatome of immature/growing/axotomized axons or axonal translatome of juvenile mice. These findings are quite unexpected and provide strong evidence that axons are capable of altering their translatome after fear conditioning. Lastly, these and recent findings by others are challenging the notion that adult axons are not capable of producing proteins locally.

On the technical side, it can be argued that TRAP-RNAseq and EM have been demonstrated in numerous publications and lacks novelty. However, as sequencing technologies and gene ontology algorithms are improving, it is necessary to i) compare and validate earlier works and to ii) establish and annotate transcriptomes/translatomes from different regions of the brain. Such works are important in establishing a methodology to study transcriptome-wide local gene expression in the context of behavioral paradigms-providing a physiological link to axonal translation.

The writing is unclear and needs to be stronger:

– While it is understandable that figures from transcriptome-wide studies are information-rich and overwhelming in the amount of data, it does not mean that the results should be annotated sloppily, left unclear or unexplained.

• Subsection “Isolation of the adult axonal translatome”, third paragraph: when explaining differential gene expression (DGE) analyses, the paragraph seems deliberately confusing by comparing two groups (translatome vs transcriptome) of two (axons vs cortex) by another set of two (experimental vs control) which all becomes undecipherable in Supplementary file 2. Should there be 16 sets of results?

• Subsection “Isolation of the adult axonal translatome”, fourth paragraph: it would be more intuitive if the selection process as noted in Figure 3—figure supplement 2A would go from a large population to small (transcriptome>>TRAP), even if the experimental work flow does not mirror that.

• If showing a number or a fraction of a population of genes, include (%) or vice versa as it is not clear which figures the numbers or percentages were taken from (subsections: “Isolation of the adult axonal translatome”, last paragraph; “Opposite learning effects in axons and cortex”, first paragraph; “Transcript-level correspondence of axonal and cortical mRNA”.

– The organizational structure of the manuscript should be changed:

• The EM data/Figures 1,2 should go after the TRAP results as supporting the TRAP outcome rather than before. The EM polysome data segues nicely from the TRAP data as there was very little explanation on why axons exhibited increased translational machinery-related mRNAs.

*Reviewer #2:*

In this provocative study, Ostroff et al. report that cortical axons projecting to the lateral amygdala in the adult rat brain contain translation machinery and mRNA translation seems to be regulated after a manipulation expected to induce associative memory in rats. Intriguingly, using TRAP with RNAseq they identified a number of mRNA transcripts that were upregulated and downregulated in axons, whereas the opposite effect was observed in cortex, suggesting a coordinated regulation between these two compartments. If true, this study represents the first demonstration for learning-regulated axonal translation in the adult mammalian brain. The relevance of this local translation to learning is assumed but never tested. What proteins are synthesized and degraded remain unknown. There is growing evidence for presynaptic protein synthesis in the adult mammalian brain and the authors seem to suggest that axonal mRNA translation could be linked to presynaptic plasticity but they do not directly address this point (not even in Discussion). While the findings are potentially significant, there are a number of claims that are not fully supported by the results.

1) The authors claim axonal protein translation occurs during memory consolidation. They also state that long-term memory formation requires de novo translation during a critical period of several hours after training and that they sacrificed animals during this time window. However, as indicated in Figure 3A measurements took place 90 min post training. Thus, the findings may not be related to memory consolidation but presynaptic protein synthesis that occurs before memory consolidation. Moreover, the authors claim axonal translation is triggered by associative learning in animals receiving paired CS+US compared to a naïve animal receiving no stimulation. To claim that associative learning induced axonal translation, control animals should receive unpaired CS+US.

2) Could the authors demonstrate learning-induced translation by measuring phospho eIF4E or S6 kinase in axons? Also, what percentage of axonal ribosomes are actively translating?

3) In Figure 5, the authors attempt to provide further evidence of mRNA localization in axons in vivo. While these data are important, the resolution that can be achieved with a laser scanning confocal microscope is only 120 nm in the xy plane, and considerably worse in the z-plane. At this distance, it is nearly impossible to argue that any FISH probes are actually contained within the axon and not in somas, dendrites, glia or anything else in the surrounding tissue. In order to convince the reader that these mRNAs are specific to axons, perhaps they could immunolabel for MAP2 and demonstrate that the target mRNAs are more often colocalized with SMI312 compared with dendritic MAP2. Alternatively, a cell-specific or higher resolution approach is needed. Lastly, the rationale for choosing mRNAs that are downregulated during learning is a bit unclear. Why not look at mRNAs upregulated under control conditions?

4) How specific are eukaryotic initiation factor antibodies? To strengthen the authors' claim for axonal translation machinery it is important to report the extent of labeling for another marker such as a ribosomal protein.

5) There seems to be a disconnect between presynaptic LTP that is supposedly induced by training and the reduction of synaptic transcripts. How do the authors explain that synapses are strengthened while synaptic proteins (e.g. Rab3A, others) are downregulated?

*Reviewer #3:*

This manuscript investigates the question of whether translation in the presynaptic compartment in the mammalian cortex is altered by learning. The authors provide immuno EM evidence that axons in the rat amygdala contain translation-associated machinery and use TRAP and RNASeq to characterise the ribosome-bound mRNAs in cortical axons projecting to the amygdala under different learning paradigms. They report that over 1200 mRNAs are regulated during the consolidation of associative memory. Mitochondrial and translation related genes, particularly, were upregulated whereas other genes such as synaptic, cytoskeletal and myelin-related, were downregulated. Also, of interest, opposite up/down gene regulation occurred in the cortex. The authors conclude that learning-regulated translation changes occur in axons of the forebrain and suggest that they may be widespread and important for learning.

This is an interesting study that reports novel findings. Previous work has taken a genome-wide approach to investigating the presynaptic translatome during learning but this has not been done for the presynaptic compartment. Indeed, axonal translation has historically been a controversial area and, although it is now broadly accepted, the important question of whether there are presynaptic changes in local translation associated with plasticity and learning in adult axons has not been investigated. The evidence presented in the manuscript is compelling and the data analysis is careful and thorough. The manuscript is written clearly and well and care is taken to give the background and rationale. Overall, this is an excellent study that adds strong in vivo evidence that local translation occurs in adult CNS axons and, further, suggests a potentially important functional role during learning. In addition, the RNASeq datasets will serve as an important resource for learning-regulated axonal translation. I have a few comments (see below) that may help improve the manuscript.

1) An unexpected and interesting finding in this study is the opposite learning effects in the axonal translatome and the cortex translatome. Although the authors suggest the possibility that compartment-specific translation is coordinated within the cell, the interpretation of this finding is not straightforward because it is not clear what the cortex translatome represents. It would be useful to see an estimate from previous literature or from additional experiments (e.g. staining for markers) indicating which types of cells express the tagged ribosome in the cortex sample and whether the neurons projecting axons to the amygdala are predominant among them.

2) In terms of the filtering method, it is confusing to use the words "axonal translatome" for the gene set that consists of two differently filtered groups: 1) the genes enriched in the axons versus the cortex and 2) those changed by the training. Their filtering excludes a lot of axonally translated genes whose translation levels are not changed by the training and not significantly higher than the translation in the cortex. Why don't the authors filter genes simply by TRAPed vs input (transcriptome) or by TRAPed vs YFP control in the basal (untrained) condition? In this context, the statement that "The majority of genes in the translatome (75%) were regulated by learning" is misleading as these regulated genes were enriched by the filtering.

[Editors’ note: this article was subsequently rejected after discussions between the reviewers, but the authors were invited to resubmit.]

Thank you for submitting your work entitled "The translatome of adult cortical axons is regulated by learning in vivo" for consideration by *eLife*. Your article has been reviewed by two peer reviewers, and the evaluation has been overseen by a Reviewing Editor and a Senior Editor. The following individuals involved in review of your submission have agreed to reveal their identity: Pablo E Castillo (Reviewer #2); Christine E Holt (Reviewer #3).

Our decision has been reached after consultation between the reviewers. Based on these discussions and the individual reviews below, we regret to inform you that your work will not be considered for publication in *eLife* in its current form. Both reviewers cite serious enough concerns to obviate publication that were not addressed in the revision. These concerns focus on the lack of proper controls and interpretation of the data. Their detailed responses are below.

The reviewers still feel the work has merit and if you feel that you can address the concerns with additional data, we will consider it as a new submission. However you may wish to transfer or submit to another journal. eLfe feels strongly that authors should not do extensive new work if they do not agree with the reviewers. So if you agree, we would be happy to see the manuscript back. If you disagree, we would strongly urge you to submit elsewhere.

Reviewer #2:

The authors have addressed some of my concerns but important issues remain unresolved.

I still think that better control experiments are required in order to claim that learning regulates the axonal translatome. To my suggestion of testing unpaired CS+US, the authors responded this protocol triggers a different form of learning which is associated with mechanistically unknown forms of synaptic plasticity. Because changes in the translatome can occur by neuronal activity triggered by CS (or US), not necessarily learning, naïve animals are not a proper control. At present, the authors' findings cannot distinguish between learning and neuronal activity induced by CS or US. Unless the authors use proper controls (e.g. CS or US, other protocol that include similar neuronal activity that paired CS+US) and unequivocally demonstrate that learning, but not neuronal activity alone, is sufficient to change the presynaptic translatome, they should remove learning as a trigger of the observed changes in the translatome. Learning-induced regulation of the translatome is presumably the most novel component of this study and therefore, much more convincing evidence is required.

The authors acknowledged that eIF antibodies may not be that specific. They explain that these essential factors cannot be genetically removed and therefore, the antibody specificity cannot be validated. This is precisely why I requested they should use a different antibody, e.g. against a ribosomal protein. The authors responded they immunolabled the ribosomal protein S6 and even observed axonal labelling with "all these factors/proteins". However, no immunoEM image labelling presynaptic/axonal rpS6 is provided.

To my request that Figure 2 should include widefield immunolabeling of YFP in lateral amygdala, the authors responded their "entire stock of virus was used" and that they "are unable to generate a new batch of tissue for this purpose". I am not persuaded by this response.

New results reported in Figure 6 are interesting but must include some quantification -e.g. extent of background labelling v. real signal. All the reader sees is some colocalization examples using poly-A RNA rather than individual mRNAs. The difference between the commercial FISH system and the "traditional FISH protocol" is not obvious and should be better clarified in the text. Lastly, the calibration bar in Figure 6 is wrong (5 μm but not 5 nm).

Reviewer #3:

Although I still generally support the publication of this manuscript, I found the authors' responses to my comments disappointing.

Their answer to point 1 is cursory and, except for adding a reference indicating that the lentivirus is selective for excitatory neurons, they do not attempt to clarify further what the cortex translatome represents.

To point 2, they do not appear to have tried to address the comment that their statement "The majority of genes in the translatome (75%) were regulated by learning" is misleading as these regulated genes were enriched by the filtering. The text remains basically the same ("The majority of genes in the translatome (1647 of 2185 or 75%) were regulated by learning" without a qualification or modification.

To the point about clarifying their use of the term the 'translatome', the authors need to be more explicit about the fact that they are looking only at ribosome-bound mRNAs and acknowledge that they have not tested the true percentage of mRNAs being translated as they have not performed standard methods to look at this (ribosome foot-printing or ribosome run-off). This distinction is important in the field and should be included for accuracy. It is not sufficient to dismiss it by saying it is 'unwieldy' to use a more accurate term. They simply need to accurately define their use of the term 'translatome' here and acknowledge that they do not know what fraction of their ribosome-bound mRNAs are being translated, if any.

Related to this point is the troubling statement in the rebuttal that eYFP itself binds to mRNA to some extent. If this is the case then it seems all the more important to compare TRAP vs YFP. Since they are not using a knock-in RP model, it also raises the question of whether the exogenously expressed eYFP-Rpl10a actually incorporates into ribosomes in their system. It is possible that the mRNA changes they detect with YFP-pulldown reflect mRNAs binding to extraribosomal YFP-Rpl10a and do not, in fact, report translation. Some serious consideration of this point should be included.

I also do not understand their reasoning "Because of the lack of consensus on what to expect in an axonal translatome and the longstanding dogma that it should not exist, we chose to focus on transcripts that we had the highest confidence were truly axonal." What factors determined their confidence?

[Editors’ note: what now follows is the decision letter after the authors submitted for further consideration.]

Thank you for resubmitting your work entitled "Axon TRAP reveals learning-induced alterations in cortical axonal mRNAs in the lateral amgydala" for further consideration at *eLife*. Your revised article has been favorably evaluated by Eve Marder as the Senior Editor, a Reviewing Editor, and two reviewers.

The manuscript has been improved but there are some remaining issues that need to be addressed before acceptance, as outlined by reviewer 2.

In short, the reviewer feels that the definition of learning needs to be clarified further in the text and some additional information provided.

Please incorporate some appropriate qualifications on the interpretations of the data into the text to address these concerns.

Reviewer #1:

The authors have answered my recent comments in a satisfactory way. They should be commended on an excellent piece of work that will have a significant impact in the fields of learning and axonal translation.

Reviewer #2:

The authors have addressed some of my concerns and the manuscript has improved, but key weaknesses remain. Although the word learning comes up 47 times in the text, the experimental evidence that learning -rather than neuronal activity- regulates the axonal translatome remains extremely weak. The authors point that "one group of rats experienced some kind of learning" is well-taken, however it does not justify the use of the phrases like "learning-regulated", which is found ubiquitously throughout the text. As I have mentioned before, the authors cannot distinguish between translatome changes that result from associative learning v. changes resulting from the foot shock or tone. Therefore, the most that they can reasonably claim is that in vivo neuronal activity (driven by specific learning paradigms) results in changes in the axonal translatome.

The fact that the authors do not observe eYFP labeling in the lateral amygdala is a bit concerning, presumably with confocal microscopy axons in the TE containing eYFP should be visible. That the authors do see eYFP labeling in axons with EM is encouraging, however it does not provide any index of the extent of the overexpression of RPL10a-eYFP construct. How much expression is needed to get sufficient mRNA? Is the amount of RPL10A-eYFP in axons consistent with the EM showing a few polyribosomes or does overexpression drive more RPL10A into the axons than is normal? Moreover, the authors do not report the amount of mRNA that was isolated, therefore it is hard to know whether the RNA isolated is of sufficient quality and quantity for subsequent RNAseq analysis.

---

## [Author Response]

Essential revisions:While the reviewers find the work interesting and important, they feel that there should be more evidence that learning leads to local translation and subsequent presynaptic plasticity: that localization of mRNAs and their translation in axons can be related to memory consolidation. They suggest higher resolution imaging and use of more specific antibodies as possible approaches. In particular, reviewer 2 suggests an additional control and would like higher resolution evidence that the mRNAs are actually in axons. Additional suggestions include using translation indicators (e.g. antibodies for phosphorylated initiation factors). Reviewer 3 has suggestions concerning how the data was analyzed.Reviewer #1:In the manuscript submitted by Ostroff et al., the authors describe the identity of axonal translatome within the lateral amygdala following auditory fear condition. Intriguingly, the results show an increase in the synthesis of mitochondrial and translation machinery-related proteins after learning. Moreover, the selective translation of such genes was most similar to the translatome of immature/growing/axotomized axons or axonal translatome of juvenile mice. These findings are quite unexpected and provide strong evidence that axons are capable of altering their translatome after fear conditioning. Lastly, these and recent findings by others are challenging the notion that adult axons are not capable of producing proteins locally.On the technical side, it can be argued that TRAP-RNAseq and EM have been demonstrated in numerous publications and lacks novelty. However, as sequencing technologies and gene ontology algorithms are improving, it is necessary to i) compare and validate earlier works and to ii) establish and annotate transcriptomes/translatomes from different regions of the brain. Such works are important in establishing a methodology to study transcriptome-wide local gene expression in the context of behavioral paradigms-providing a physiological link to axonal translation.The writing is unclear and needs to be stronger:– While it is understandable that figures from transcriptome-wide studies are information-rich and overwhelming in the amount of data, it does not mean that the results should be annotated sloppily, left unclear or unexplained.• Subsection “Isolation of the adult axonal translatome”, third paragraph: when explaining differential gene expression (DGE) analyses, the paragraph seems deliberately confusing by comparing two groups (translatome vs transcriptome) of two (axons vs cortex) by another set of two (experimental vs control) which all becomes undecipherable in Supplementary file 2. Should there be 16 sets of results?

There are a total of 12 sets of DGE results, but we did not intend for the text to be confusing. This has been rewritten for clarity.

• Subsection “Isolation of the adult axonal translatome”, fourth paragraph: it would be more intuitive if the selection process as noted in Figure 3—figure supplement 2A would go from a large population to small (transcriptome>>TRAP), even if the experimental work flow does not mirror that.

The steps shown were conducted after each DGE analysis, and we have clarified that in the legend. We have rearranged the flow chart as suggested, in order of the number of transcripts filtered out at each step

• If showing a number or a fraction of a population of genes, include (%) or vice versa as it is not clear which figures the numbers or percentages were taken from (subsections: “Isolation of the adult axonal translatome”, last paragraph; “Opposite learning effects in axons and cortex”, first paragraph; “Transcript-level correspondence of axonal and cortical mRNA”.

We have clarified these statements.

– The organizational structure of the manuscript should be changed:• The EM data/Figures 1, 2 should go after the TRAP results as supporting the TRAP outcome rather than before. The EM polysome data segues nicely from the TRAP data as there was very little explanation on why axons exhibited increased translational machinery-related mRNAs.

We certainly see the reviewer’s point, although we respectfully disagree. In reality, the TRAP experiment was undertaken to confirm and explore our unexpected observations of polysomes and immunolabeling of translation machinery in axons by EM. The order of the manuscript does not necessarily need to reflect the initial logic of the project, of course, but here we feel that direct visualization of this elusive phenomenon sets the stage for deeper exploration in the form of sequencing data.

Reviewer #2:In this provocative study, Ostroff et al. report that cortical axons projecting to the lateral amygdala in the adult rat brain contain translation machinery and mRNA translation seems to be regulated after a manipulation expected to induce associative memory in rats. Intriguingly, using TRAP with RNAseq they identified a number of mRNA transcripts that were upregulated and downregulated in axons, whereas the opposite effect was observed in cortex, suggesting a coordinated regulation between these two compartments. If true, this study represents the first demonstration for learning-regulated axonal translation in the adult mammalian brain. The relevance of this local translation to learning is assumed but never tested. What proteins are synthesized and degraded remain unknown. There is growing evidence for presynaptic protein synthesis in the adult mammalian brain and the authors seem to suggest that axonal mRNA translation could be linked to presynaptic plasticity but they do not directly address this point (not even in Discussion). While the findings are potentially significant, there are a number of claims that are not fully supported by the results.

*1) The authors claim axonal protein translation occurs during memory consolidation. They also state that long-term memory formation requires* de novo *translation during a critical period of several hours after training and that they sacrificed animals during this time window. However, as indicated in Figure 3A measurements took place 90 min post training. Thus, the findings may not be related to memory consolidation but presynaptic protein synthesis that occurs before memory consolidation.*

The process of memory consolidation is generally characterized by successive waves of learning-induced gene expression cascades that begin within minutes and last for at least several hours. There is a substantial literature on translation of newly transcribed and pre-existing mRNAs over minutes to hours, and on transport of mRNA into dendrites over this period. In previous work, we observed increased polyribosomes in dendrites 30 minutes after learning. Distal axons are an order of magnitude farther from the soma than distal dendrites and reported rates of mRNA transport in neurons are highly variable. We chose a later time point, still well within the active consolidation phase, on the hypothesis that mRNAs may be trafficked from the soma and would need more time to travel to distal axons.

Moreover, the authors claim axonal translation is triggered by associative learning in animals receiving paired CS+US compared to a naïve animal receiving no stimulation. To claim that associative learning induced axonal translation, control animals should receive unpaired CS+US.

We are indeed claiming that our trained group underwent associative learning and our control group did not, but we are not claiming that our observations are specific to the excitatory aversive association. Unpaired training produces conditioned inhibition, specifically safety learning in this case, a form of associative learning in which the CS acquires the ability to suppress fear and anxiety. We have found structural plasticity in the LA as a result of the unpaired protocol relative to the paired training and box control protocols presented here, and published several behavioral controls confirming conditioned inhibition (Ostroff et al., 2010). The unpaired protocol also results in reduced tone-evoked synaptic responses in the LA, opposite to the changes induced by paired training (Rogan et al., 1997 Nature; Rogan et al., 2005). The synaptic plasticity mechanisms of conditioned inhibition are unknown, so instead of comparing two related types of associative learning to each other, we chose to expose both groups to a familiar context and train only one.

2) Could the authors demonstrate learning-induced translation by measuring phospho eIF4E or S6 kinase in axons? Also, what percentage of axonal ribosomes are actively translating?

This a great suggestion, but we have never succeeded in specifically immunolabeling phosphorylated proteins in fixed tissue, presumably due to continued activity of phosphatases during and after aldehyde fixation or to the lower number of epitopes specific to a phosphorylated state. We do not know of a straightforward method for assessing the temporal dynamics of active translation in this paradigm. The presence of initiation factors in axons suggests that translation may be initiated in at least some transcripts in the axons, as opposed to 100% of axonal ribosome-bound transcripts having undergone initiation in the soma. In addition, it is possible that transcripts TRAPed in the control axons, but not trained axons, were translated locally before being degraded. Experiments to directly assess in vivorates of translation of ribosome-bound transcripts would require extensive protocol development that is beyond the scope of this study.

*3) In Figure 5, the authors attempt to provide further evidence of mRNA localization in axons* in vivo*. While these data are important, the resolution that can be achieved with a laser scanning confocal microscope is only 120 nm in the xy plane, and considerably worse in the z-plane. At this distance, it is nearly impossible to argue that any FISH probes are actually contained within the axon and not in somas, dendrites, glia or anything else in the surrounding tissue. In order to convince the reader that these mRNAs are specific to axons, perhaps they could immunolabel for MAP2 and demonstrate that the target mRNAs are more often colocalized with SMI312 compared with dendritic MAP2. Alternatively, a cell-specific or higher resolution approach is needed. Lastly, the rationale for choosing mRNAs that are downregulated during learning is a bit unclear. Why not look at mRNAs upregulated under control conditions?*

We agree that the resolution of confocal microscopy is a concern. The transcripts we chose are present in the local cell bodies and potentially in local dendrites, so even if we were to find substantial colocalization with MAP2 that would not preclude axonal localization. By EM, the smallest axon diameters are about 100 nm. The EM preparations we have used here represent at least 20% tissue shrinkage, and substantial loss of extracellular space. In our experience with EM work in different preparations, under less harsh processing tissue components are larger and less tightly packed, and the XY resolution of confocal is sufficient to co-localize labels in axons. The Z resolution remains problematic, though. We have performed a new experiment on gently fixed, resin-embedded samples sectioned at 100 nm. In this preparation we were able to see an oligo(dT) probe co-localized with smi312, further indicating the presence of poly-A mRNA in axons.

Our reasoning in choosing those particular transcripts was that transcripts upregulated by learning are likely part of a complex gene expression cascade and their translation levels may fluctuate during consolidation. Downregulated genes may be more likely to be constitutively translated under control conditions.

4) How specific are eukaryotic initiation factor antibodies? To strengthen the authors' claim for axonal translation machinery it is important to report the extent of labeling for another marker such as a ribosomal protein.

We agree that antibody specificity is very important. The usual control for antibody specificity, acute removal of the antigen from the tissue such as by conditional genetic knockout, is not available for such essential target molecules. Because we could not remove, for instance, eIF4E from the tissue to confirm that the axonal labelling was not an artifact, we immunolabeled several other translation factors, as well as a ribosomal protein (rpS6). We observed axonal labelling of all of these factors/proteins

5) There seems to be a disconnect between presynaptic LTP that is supposedly induced by training and the reduction of synaptic transcripts. How do the authors explain that synapses are strengthened while synaptic proteins (e.g. Rab3A, others) are downregulated?

Our data reflect the levels of ribosome-bound transcripts, not protein or dormant transcripts. One interpretation of our data is that the loss of ribosome-bound transcripts is due to translation and thus accumulation of the protein (and degradation of the transcript).

Reviewer #3:This manuscript investigates the question of whether translation in the presynaptic compartment in the mammalian cortex is altered by learning. The authors provide immuno EM evidence that axons in the rat amygdala contain translation-associated machinery and use TRAP and RNASeq to characterise the ribosome-bound mRNAs in cortical axons projecting to the amygdala under different learning paradigms. They report that over 1200 mRNAs are regulated during the consolidation of associative memory. Mitochondrial and translation related genes, particularly, were upregulated whereas other genes such as synaptic, cytoskeletal and myelin-related, were downregulated. Also, of interest, opposite up/down gene regulation occurred in the cortex. The authors conclude that learning-regulated translation changes occur in axons of the forebrain and suggest that they may be widespread and important for learning.
*This is an interesting study that reports novel findings. Previous work has taken a genome-wide approach to investigating the presynaptic translatome during learning but this has not been done for the presynaptic compartment. Indeed, axonal translation has historically been a controversial area and, although it is now broadly accepted, the important question of whether there are presynaptic changes in local translation associated with plasticity and learning in adult axons has not been investigated. The evidence presented in the manuscript is compelling and the data analysis is careful and thorough. The manuscript is written clearly and well and care is taken to give the background and rationale. Overall, this is an excellent study that adds strong* in vivo evidence that local translation occurs in adult CNS axons and, further, suggests a potentially important functional role during learning. In addition, the RNASeq datasets will serve as an important resource for learning-regulated axonal translation. I have a few comments (see below) that may help improve the manuscript.1) An unexpected and interesting finding in this study is the opposite learning effects in the axonal translatome and the cortex translatome. Although the authors suggest the possibility that compartment-specific translation is coordinated within the cell, the interpretation of this finding is not straightforward because it is not clear what the cortex translatome represents. It would be useful to see an estimate from previous literature or from additional experiments (e.g. staining for markers) indicating which types of cells express the tagged ribosome in the cortex sample and whether the neurons projecting axons to the amygdala are predominant among them.

Unfortunately, none of our viral TRAP reagent remains after our experiments so we are unable to repeat the transfections to double stain for markers. Lentivirus has been shown to be selective for excitatory neurons, however, and we have added a reference to this.

2) In terms of the filtering method, it is confusing to use the words "axonal translatome" for the gene set that consists of two differently filtered groups: 1) the genes enriched in the axons versus the cortex and 2) those changed by the training. Their filtering excludes a lot of axonally translated genes whose translation levels are not changed by the training and not significantly higher than the translation in the cortex. Why don't the authors filter genes simply by TRAPed vs input (transcriptome) or by TRAPed vs YFP control in the basal (untrained) condition? In this context, the statement that "The majority of genes in the translatome (75%) were regulated by learning" is misleading as these regulated genes were enriched by the filtering.

We took the most stringent approach we could think of to define our “axonal translatome,” which we agree almost certainly excludes many transcripts that actually are in axons. Because of the lack of consensus on what to expect in an axonal translatome and the longstanding dogma that it should not exist, we chose to focus on transcripts that we had the highest confidence were truly axonal.

We did not do a direct DGE analysis between our TRAP and YFP data for two reasons: First, the TRAP and YFP experiments were not conducted together as a single batch. Between the need to pool animals for TRAP-seq and our rigorous tissue collection approach, we were only able to collect samples from one set of trained and control animals per experiment. Second, we cannot exclude the possibility that eYFP itself binds RNA to some extent, and we did in fact find some mRNAs enriched in our YFP pulldowns versus the tissue transcriptome. Because the YFP was in the transfected TE3 cells and their axons, YFP pulldowns might enrich for mRNA from these cells and potentially result in false negatives depending on the fraction of a given transcript that is ribosome-bound in the axons.

[Editors’ note: this article was subsequently rejected after discussions between the reviewers, but the authors were invited to resubmit.]

The reviewers still feel the work has merit and if you feel that you can address the concerns with additional data, we will consider it as a new submission. However you may wish to transfer or submit to another journal. eLfe feels strongly that authors should not do extensive new work if they do not agree with the reviewers. So if you agree, we would be happy to see the manuscript back. If you disagree, we would strongly urge you to submit elsewhere.Reviewer #2:The authors have addressed some of my concerns but important issues remain unresolved.I still think that better control experiments are required in order to claim that learning regulates the axonal translatome. To my suggestion of testing unpaired CS+US, the authors responded this protocol triggers a different form of learning which is associated with mechanistically unknown forms of synaptic plasticity. Because changes in the translatome can occur by neuronal activity triggered by CS (or US), not necessarily learning, naïve animals are not a proper control. At present, the authors' findings cannot distinguish between learning and neuronal activity induced by CS or US. Unless the authors use proper controls (e.g. CS or US, other protocol that include similar neuronal activity that paired CS+US) and unequivocally demonstrate that learning, but not neuronal activity alone, is sufficient to change the presynaptic translatome, they should remove learning as a trigger of the observed changes in the translatome. Learning-induced regulation of the translatome is presumably the most novel component of this study and therefore, much more convincing evidence is required.

We respectfully disagree with the reviewer on the appropriate control for this experiment, to which we gave much consideration. First, it should be noted that the rats were not naïve because they were handled and placed in the conditioning box during habituation as well as during training for the same duration as the learning group. As we previously stated and as the reviewer agrees, exposing the animals to the tone-alone or shock-alone would be manifest as novel sensory experiences and likely engage neuronal populations within the amygdala, especially in the case of an innately aversive US. The reviewer is correct in that if we had used tone-alone, shock-alone, or unpaired, we could say that any differences in translation we observed were due to threat conditioning and captured the tone-shock contingency. What we can say is that one group of rats experienced some kind of learning (whether the CS, US, or the CS+US association) and the other group sat in a familiar environment and learned nothing new. We intentionally used “learning” in the title of the manuscript because we felt using the term “threat conditioning” might be misleading, as the reviewer noted. We also used the term training throughout the manuscript to describe precisely what was triggering the changes we observed. If the reviewer feels strongly that we should not use the term learning in the title, we would be willing to change it to something mutually agreeable.

The authors acknowledged that eIF antibodies may not be that specific. They explain that these essential factors cannot be genetically removed and therefore, the antibody specificity cannot be validated. This is precisely why I requested they should use a different antibody, e.g. against a ribosomal protein. The authors responded they immunolabled the ribosomal protein S6 and even observed axonal labelling with "all these factors/proteins". However, no immunoEM image labelling presynaptic/axonal rpS6 is provided.

We actually had presented an immunoEM image of rpS6 labelling in both the original and revised manuscript in Figure 1F. We apologize for not pointing this out more clearly in our previous response to the reviewer’s comment.

To my request that Figure 2 should include widefield immunolabeling of YFP in lateral amygdala, the authors responded their "entire stock of virus was used" and that they "are unable to generate a new batch of tissue for this purpose". I am not persuaded by this response.

In our previous response, we neglected to say that we did not observe clear eYFP in the lateral amygdala by widefield imaging before we processed our samples for immunoEM, which permitted us to unambiguously observe eYFP immunolabeling in the axons. We now state this clearly in the revised manuscript.

New results reported in Figure 6 are interesting but must include some quantification -e.g. extent of background labeling v. real signal. All the reader sees is some colocalization examples using poly-A RNA rather than individual mRNAs. The difference between the commercial FISH system and the "traditional FISH protocol" is not obvious and should be better clarified in the text. Lastly, the calibration bar in Figure 6 is wrong (5 μm but not 5 nm).

We now included quantification for Figure 6. We also have clarified the difference between the commercial FISH system and traditional FISH protocol. We also have corrected the calibration bar in Figure 6. We thank the reviewer for catching this error.

Reviewer #3:Although I still generally support the publication of this manuscript, I found the authors' responses to my comments disappointing.

We are pleased that the reviewer supports publication and apologize for the way we responded to the previous critique of the manuscript.

Their answer to point 1 is cursory and, except for adding a reference indicating that the lentivirus is selective for excitatory neurons, they do not attempt to clarify further what the cortex translatome represents.

We did not intend for our answer to be cursory and should have been more clear in our previous response. We do say: “Although we refer to these samples as cortex and axons, the cortex samples also contain proximal axon segments, myelinated segments that pass through the dorsal portion of the external capsule, as well as intrinsic projections and corticocortical projections terminating in the adjacent areas of TE1 and perirhinal cortex.”

To point 2, they do not appear to have tried to address the comment that their statement "The majority of genes in the translatome (75%) were regulated by learning" is misleading as these regulated genes were enriched by the filtering. The text remains basically the same ("The majority of genes in the translatome (1647 of 2185 or 75%) were regulated by learning" without a qualification or modification.

We apologize for the lack of clarity in our previous response. We state in the text: “It is important to note that although we are using the term “translatome” to refer to the strangely selected subset of genes we used for analysis, the actual population of axonal mRNAs is almost certainly larger.”

We understand that the reviewer disagrees with our filtering of the results, but we do demonstrate in Figure 3—figure supplement 3A that the filtering did not bias the analysis we present. The filtering step is why we have confidence in the set of transcripts that were analyzed. We removed every source of experimental background that we could and we openly state that we are sure that there are false negatives. All of the raw data and analysis are publicly available, so the unfiltered lists are available to anyone who wants to analyze them and use a less conservative definition of the term “axonal”. As we now discuss, the mRNAs isolated from axons overexpressing eYFP are likely enriched in axonal transcripts without bias for ribosome binding, and may themselves be of interest.

To the point about clarifying their use of the term the 'translatome', the authors need to be more explicit about the fact that they are looking only at ribosome-bound mRNAs and acknowledge that they have not tested the true percentage of mRNAs being translated as they have not performed standard methods to look at this (ribosome foot-printing or ribosome run-off). This distinction is important in the field and should be included for accuracy. It is not sufficient to dismiss it by saying it is 'unwieldy' to use a more accurate term. They simply need to accurately define their use of the term 'translatome' here and acknowledge that they do not know what fraction of their ribosome-bound mRNAs are being translated, if any.

We agree with the reviewer. We again apologize for our response and for not being more explicit about the use of the term “translatome”. The reviewer is correct that using TRAP we cannot know whether the ribosomes on a given mRNA are in the process of active elongation at the moment of tissue harvest, whether they are in a stalled state undergoing transport, or whether they are awaiting reactivation of elongation. Because the TRAP method does not capture mRNAs that are not ribosome-bound, it excludes any masked mRNAs and therefore TRAPed mRNAs do not represent an axonal transcriptome. We have softened the language in the Introduction, and expanded the Discussion to explicitly address what our data represent in the context of translation regulation.

Related to this point is the troubling statement in the rebuttal that eYFP itself binds to mRNA to some extent. If this is the case then it seems all the more important to compare TRAP vs YFP. Since they are not using a knock-in RP model, it also raises the question of whether the exogenously expressed eYFP-Rpl10a actually incorporates into ribosomes in their system. It is possible that the mRNA changes they detect with YFP-pulldown reflect mRNAs binding to extraribosomal YFP-Rpl10a and do not, in fact, report translation. Some serious consideration of this point should be included.

We understand the reviewer’s concern. We acknowledge that with TRAP some RNAs will bind, whether it’s specific or non-specific, to the eYFP tag, just as some RNAs will bind to the HA tag with ribo-tag. Even if RNA does bind to eYFP in the amygdala, under our experimental conditions it is still axonal. We also have added text stating that the YFP pulldown could reflect mRNAs binding extraribosomal eYFP-Rpl10a. However, it should be noted and as expected, that very few genes showed learning-induced changes in the eYFP data set (Figure 3—figure supplement 2), and this data set showed no meaningful functional enrichment and no functional overlap with the TRAPed set (Figure 3—figure supplement 2).

[Editors’ note: what now follows is the decision letter after the authors submitted for further consideration.]

Reviewer #1:The authors have answered my recent comments in a satisfactory way. They should be commended on an excellent piece of work that will have a significant impact in the fields of learning and axonal translation.

We are pleased that the reviewer finds the work of value to the field.

Reviewer #2:

*The authors have addressed some of my concerns and the manuscript has improved, but key weaknesses remain. Although the word learning comes up 47 times in the text, the experimental evidence that learning -rather than neuronal activity- regulates the axonal translatome remains extremely weak. The authors point that "one group of rats experienced some kind of learning" is well-taken, however it does not justify the use of the phrases like "learning-regulated", which is found ubiquitously throughout the text. As I have mentioned before, the authors cannot distinguish between translatome changes that result from associative learning v. changes resulting from the foot shock or tone. Therefore, the most that they can reasonably claim is that* in vivo *neuronal activity (driven by specific learning paradigms) results in changes in the axonal translatome.*

As we’ve noted, we gave very careful consideration to the choice of training and control protocols before beginning the project, and we have added a paragraph to the discussion explaining how the choice was made. The reviewer is correct that our experimental design does not isolate the effects of excitatory associative learning between the tone and the shock, nor was it intended to. The reviewer’s concern, as we understand it, is that our results could be induced by neural activity downstream of the hair cell and nociceptor stimulation delivered by the tone and shock, as opposed to the concomitant learning. All other things being equal, learning can be no more easily separated from that neural activity than the activity can be separated from activation of the peripheral mechanoreceptors. This makes the choice of controls a complex one.

The training protocol that we used – paired tones and shocks given in a familiar context to animals naïve to the stimuli – induces strong excitatory associative learning. The excitatory association is often controlled for by presenting explicitly unpaired tones and shocks, which produces inhibitory associative learning to the tone and excitatory associative learning to the training context. Novel tones alone produce latent inhibition, a form of non-associative learning, along with remapping of auditory receptive fields and decreased neural responses to the tone. Well habituated tones can be presented without inducing additional learning, but the same degree of habituation would need to be given to all subjects and the resulting latent inhibition would impair learning in the trained group. Novel shocks produce associative learning to the stimulus with the strongest predictive value. Shocks given in a previously trained protocol produce conditioned analgesia, in which the nociceptive processing is dampened within the learning circuit. Prior shocks in a context also produce blocking, the inhibition of new associative learning in the presence of a previously learned predictor. All of these phenomena are extensively documented in decades of literature and are covered in standard behavioral textbooks, we have repeatedly observed all of them in our own experiments, and we have no reason to doubt that they represent genuine concerns.

In essence, our options were to compare different types of learning to each other (different associations or associative vs. non-associative), associative learning in a naïve subject to stimulus exposure in a habituated subject, or learning in a familiar context to exposure to a familiar context. We chose the last. Under ideal circumstances we would have run multiple control groups, even with the expense involved. However, our tissue collections were labor-intensive and timed to the minute, and to further avoid artifact and variability we collected samples from the same cohort of rats within a limited period of the light cycle. After pooling samples for sequencing, there was no practical way to run more than two experimental groups without sacrificing rigor in other aspects. If we were measuring a phenomenon whose normal parameters were well understood – neuronal membrane properties or dendritic spine density, for example – it would be an easy choice to isolate one type of learning. Having no a priori expectation of the nature of an adult axonal translatome, we chose the broader comparison of learning versus an experience of relatively low salience. This allowed us to answer simpler questions regarding the basic composition of the axonal translatome and whether it is subject to regulation by experience-induced plasticity (i.e. learning). We have avoided claiming that the effects we observed were specific to excitatory associative learning, and our use of the term “learning” refers collectively to the various changes in information processing that are the result of stimulus presentations.

The fact that the authors do not observe eYFP labeling in the lateral amygdala is a bit concerning, presumably with confocal microscopy axons in the TE containing eYFP should be visible. That the authors do see eYFP labeling in axons with EM is encouraging, however it does not provide any index of the extent of the overexpression of RPL10a-eYFP construct. How much expression is needed to get sufficient mRNA? Is the amount of RPL10A-eYFP in axons consistent with the EM showing a few polyribosomes or does overexpression drive more RPL10A into the axons than is normal? Moreover, the authors do not report the amount of mRNA that was isolated, therefore it is hard to know whether the RNA isolated is of sufficient quality and quantity for subsequent RNAseq analysis.

We did not find it particularly surprising that eYFP was not visible by light microscopy in axons. Although it was easily visible in the transfected cell bodies, where RPL10a is presumably more concentrated on rough ER, we would not necessarily expect a strong fluorescence signal to be generated from RPL10a-eYFP in axons, where the enveloping myelin, vesicle pools, and white matter can interfere with both excitation and emission of photons, especially if the molecules are not present in high concentrations. We focused on EM imaging because it is more sensitive and can unambiguously localize signal in compartments such as axons.

As we now explain in the Discussion, translation is regulated upstream of ribosome recruitment, so there is no reason to expect that translation would be significantly altered by the presence of excess ribosomal proteins. In addition, the function of overexpressed RPL10a-eYFP was evaluated by Heiman et al. in the initial report of the TRAP technique and found to be equivalent to the endogenous protein. We have clarified in the text that the scarcity of polyribosomes in LA axons is consistent with reports of translation without polyribosomes in other types of axons, and likely reflects alternate translational structures such as monosomes. All final mRNA yields and quality control measures are given in Supplementary File 1. RNA was quantified after amplification, since the raw yield after the IP step is a function of the number of cells infected by the virus, the number of infected cells that project axons to the amygdala, and the amount of white matter collected during the dissection.